# Collective Robustness Certificates: Exploiting Interdependence in Graph Neural Networks

**Jan Schuchardt, Aleksandar Bojchevski, Johannes Gasteiger & Stephan Günnemann**
Technical University of Munich, Germany
`{jan.schuchardt,bojchevs,j.gasteiger,guennemann}@in.tum.de`

## Abstract

In tasks like node classification, image segmentation, and named-entity recognition we have a classifier that simultaneously outputs multiple predictions (a vector of labels) based on a single input, i.e. a single graph, image, or document respectively. Existing adversarial robustness certificates consider each prediction independently and are thus overly pessimistic for such tasks. They implicitly assume that an adversary can use different perturbed inputs to attack different predictions, ignoring the fact that we have a *single* shared input. We propose the first collective robustness certificate which computes the number of predictions that are *simultaneously* guaranteed to remain stable under perturbation, i.e. cannot be attacked. We focus on Graph Neural Networks and leverage their locality property – perturbations only affect the predictions in a close neighborhood – to fuse multiple single-node certificates into a drastically stronger collective certificate. For example, on the Citeseer dataset our collective certificate for node classification increases the average number of certifiable feature perturbations from 7 to 351.

## 1 Introduction

Most classifiers are vulnerable to adversarial attacks (Akhtar & Mian, 2018; Hao-Chen et al., 2020). Slight perturbations of the data are often sufficient to manipulate their predictions. Even in scenarios where attackers are not present it is critical to ensure that models are robust since data can be noisy, incomplete, or anomalous. We study classifiers that collectively output many predictions based on a single input. This includes node classification, link prediction, molecular property prediction, image segmentation, part-of-speech tagging, named-entity recognition, and many other tasks.

Various techniques have been proposed to improve the adversarial robustness of such models. One example is adversarial training (Goodfellow et al., 2015), which has been applied to part-of-speech tagging (Han et al., 2020), semantic segmentation (Xu et al., 2020b) and node classification (Feng et al., 2019). Graph-related tasks in particular have spawned a rich assortment of techniques. These include Bayesian models (Feng et al., 2020), data-augmentation methods (Entezari et al., 2020) and various robust network architectures (Zhu et al., 2019; Geisler et al., 2020). There are also robust loss functions which either explicitly model an adversary trying to cause misclassifications (Zhou & Vorobeychik, 2020) or use regularization terms derived from robustness certificates (Zügner & Günnemann, 2019). Other methods try to detect adversarially perturbed graphs (Zhang et al., 2019; Xu et al., 2020a) or directly correct perturbations using generative models (Zhang & Ma, 2020).

However, none of these techniques provide guarantees and they can only be evaluated based on their ability to defend against known adversarial attacks. Once a technique is established, it may subsequently be defeated using novel attacks (Carlini & Wagner, 2017). We are therefore interested in deriving adversarial robustness certificates which provably guarantee that a model is robust.

In this work we focus on node classification.[1] Here, the goal is to assign a label to each node in a single (attributed) graph. Node classification can be the target of either *local* or *global* adversarial attacks. Local attacks, such as Nettack (Zügner et al., 2018; Zügner et al., 2020), attempt to alter the

---

[1]While we focus on node classification, our approach can easily be applied to other multi-output classifiers.

Figure 1: Previous certificates consider each node independently. Most nodes cannot be certified since the adversary can choose a different perturbed graph per node (left). This is impossible in practice due to mutually exclusive perturbations. Our collective certificate enforces a single perturbed graph (center). It aggregates the amount of perturbation within each receptive field and then evaluates a single-node certificate to determine whether the corresponding prediction is robust (right).

prediction of a particular node in the graph. Global attacks, as proposed by Zügner & Günnemann (2019), attempt to alter the predictions of many nodes at once. With global attacks, the attacker is constrained by the fact that all predictions are based on a *single* shared input. To successfully attack some nodes the attacker might need to insert certain edges in the graph, while for another set of nodes the same edges must not be inserted. With such mutually exclusive adversarial perturbations, the attacker is forced to make a choice and can attack only one subset of nodes (see Fig. 1).

Existing certificates (Zügner & Günnemann, 2019; Bojchevski & Günnemann, 2019; Bojchevski et al., 2020) are designed for local attacks, i.e. to certify the predictions of individual nodes. So far, there is no dedicated certificate for global attacks, i.e. to certify the predictions of many nodes at once[2]. A naïve certificate for global attacks can be constructed from existing single-node certificates as follows: One simply certifies each node's prediction independently and counts how many are guaranteed to be robust. This, however, implicitly assumes that an adversary can use different perturbed inputs to attack different predictions, ignoring the fact that we have a single shared input.

We propose a *collective robustness certificate* for global attacks that directly computes the number of *simultaneously* certifiable nodes for which we can guarantee that their predictions will not change. This certificate explicitly models that the attacker is limited to a single shared input and thus accounts for the resulting mutual exclusivity of certain attacks. Specifically, we fuse multiple single-node certificates, which we refer to as base certificates, into a drastically (and provably) stronger collective one. Our approach is independent of how the base certificates are derived, and any improvements to the base certificates directly translate to improvements to the collective certificate.

The key property which we exploit is *locality*. For example, in a $k$-layer message-passing graph neural network (Gilmer et al., 2017) the prediction for any given node depends only on the nodes in its $k$-hop neighborhood. Similarly, the predicted segment for any pixel depends only on the pixels in its receptive field, and the named-entity assigned to any word only depends on words in its surrounding.

For classifiers that satisfy locality, perturbations to one part of the graph do not affect all nodes. Adversaries are thus faced with a budget allocation problem: It might be possible to attack different subsets of nodes via perturbations to different subgraphs, but performing all perturbations at once could exceed their adversarial budget. The naïve approach discussed above ignores this, over-estimating how many nodes can be attacked. We design a simple (mixed-integer) linear program (LP) that enforces a single perturbed graph. It leverages locality by only considering the amount of perturbation within each receptive field when evaluating the single-node certificates (see Fig. 1).

We evaluate our approach on different datasets and with different base certificates. We show that incorporating locality alone is sufficient to obtain significantly better results. Our proposed certificate:

- Is the first *collective* certificate that explicitly models simultaneous attacks on multiple outputs.
- Fuses individual certificates into a provably stronger certificate by explicitly modeling *locality*.
- Is the first node classification certificate that can model not only global and local budgets, but also the number of adversary-controlled nodes, regardless of whether the base certificates support this.

---

[2]Chiang et al. (2020) certify multi-object detection, but they still treat each detected object independently.

## 2  PRELIMINARIES

**Data and models.** We define our unperturbed data as an attributed graph $\mathcal{G} = (\boldsymbol{X}, \boldsymbol{A}) \in \mathbb{G}$ with $\mathbb{G} = \{0, 1\}^{N \times D} \times \{0, 1\}^{N \times N}$, consisting of $N$ $D$-dimensional feature vectors and a directed $N \times N$ adjacency matrix. Each vertex is assigned one out of $C$ classes by a multi-output classifier $f : \mathbb{G} \mapsto \{1, \dots, C\}^N$. In the following, $f_n(\mathcal{G}) = f_n(\boldsymbol{X}, \boldsymbol{A}) = f_n$ refers to the prediction for node $n$.

**Collective threat model.** Unlike previous certificates, we model an adversary that aims to change multiple predictions at once. Let $\mathbb{B}_{\mathcal{G}} \subseteq \mathbb{G}$ be a set of admissible perturbed graphs. Given a clean graph $\mathcal{G}$, the adversary tries to find a $\mathcal{G}' \in \mathbb{B}_{\mathcal{G}}$ that maximizes the number of misclassified nodes, i.e. $\sum_{n \in \mathbb{T}} \mathbf{1}_{f_n(\mathcal{G}) \neq f_n(\mathcal{G}')}$, for some set of target nodes $\mathbb{T} \subseteq \{1, \dots, N\}$.

Following prior work (Zügner & Günnemann, 2019), we constrain the set of admissible perturbed graphs $\mathbb{B}_{\mathcal{G}}$ through global and (optionally) local constraints on the number of changed bits. Our global constraints are parameterized by scalars $r_{\boldsymbol{X}_{\mathrm{add}}}, r_{\boldsymbol{X}_{\mathrm{del}}}, r_{\boldsymbol{A}_{\mathrm{add}}}, r_{\boldsymbol{A}_{\mathrm{del}}} \in \mathbb{N}_0$. They are an upper limit on how many bits can be added ($0 \to 1$) or deleted ($1 \to 0$) when perturbing $\boldsymbol{X}$ and $\boldsymbol{A}$. Our local constraints are parameterized by vectors $\boldsymbol{r}_{\boldsymbol{X}_{\mathrm{add,loc}}}, \boldsymbol{r}_{\boldsymbol{X}_{\mathrm{del,loc}}}, \boldsymbol{r}_{\boldsymbol{A}_{\mathrm{add,loc}}}, \boldsymbol{r}_{\boldsymbol{A}_{\mathrm{del,loc}}} \in \mathbb{N}_0^N$. They are an upper limit on how many bits can be added or deleted per row of $\boldsymbol{X}$ and $\boldsymbol{A}$, i.e. how much the attributes of a particular node can change and how many incident edges can be perturbed.

Often, it is reasonable to assume that no adversary has direct control over the entire graph. Instead, a realistic attacker should only be able to perturb a small (adaptively chosen) subset of nodes. To model this, we introduce an additional parameter $\sigma \in \mathbb{N}$. For all $(\boldsymbol{X}', \boldsymbol{A}') \in \mathbb{B}_{\mathcal{G}}$, there must be a set of node indices $\mathbb{S} \subseteq \{1, \dots, N\}$ with $|\mathbb{S}| \leq \sigma$ such that for all $d \in \{1, \dots, D\}$ and $n, m \in \{1, \dots, N\}$:

$$\left( X'_{n,d} \neq X_{n,d} \implies n \in \mathbb{S} \right) \wedge \left( A'_{n,m} \neq A_{n,m} \implies n \in \mathbb{S} \vee m \in \mathbb{S} \right). \tag{1}$$

The set $\mathbb{S}$ is not fixed, but chosen by the adversary.[3] The resulting set $\mathbb{B}_{\mathcal{G}}$ is formally defined in Section B. If the global budget parameters are variable and the remaining parameters are clear from the context, we treat $\mathbb{B}_{\mathcal{G}}$ as a function $\mathbb{B}_{\mathcal{G}} : \mathbb{N}_0^4 \mapsto \mathcal{P}(\mathbb{G})$ that, given global budget parameters, returns the set of all perturbed graphs fulfilling the constraints.

**Local predictions.** Our certificate exploits the locality of predictions, i.e. the fact that predictions are only based on a subset of the input data. We characterize the receptive field of $f_n$ via an indicator vector $\boldsymbol{\psi}^{(n)} \in \{0, 1\}^N$ corresponding to rows in attribute matrix $\boldsymbol{X}$ and an indicator matrix $\boldsymbol{\Psi}^{(n)} \in \{0, 1\}^{N \times N}$ corresponding to entries in adjacency matrix $\boldsymbol{A}$. For all $(\boldsymbol{X}', \boldsymbol{A}'), (\boldsymbol{X}'', \boldsymbol{A}'') \in \mathbb{B}_{\mathcal{G}}$:

$$\sum_{m=1}^{N} \sum_{d=1}^{D} \psi_m^{(n)} \mathbf{1}_{X'_{m,d} \neq X''_{m,d}} + \sum_{i=1}^{N} \sum_{j=1}^{N} \Psi_{i,j}^{(n)} \mathbf{1}_{A'_{i,j} \neq A''_{i,j}} = 0 \implies f_n(\boldsymbol{X}', \boldsymbol{A}') = f_n(\boldsymbol{X}'', \boldsymbol{A}''). \tag{2}$$

Eq. 2 enforces that as long as all nodes and edges for which $\boldsymbol{\psi}^{(n)} = 1$ or $\boldsymbol{\Psi}^{(n)} = 1$ remain unperturbed, the prediction $f_n$ does not change. Put differently, changes to the rest of the data do not affect the prediction. Note that the adversary can alter receptive fields, e.g. add edges to enlarge them. To capture all potential alterations, $\boldsymbol{\psi}^{(n)}$ and $\boldsymbol{\Psi}^{(n)}$ correspond to all data points that influence $f_n$ under some graph in $\mathbb{B}_{\mathcal{G}}$, i.e. the union of all receptive fields achievable under the threat model.

## 3  COMPACT REPRESENTATION OF BASE CERTIFICATES

Before deriving our collective certificate, we define a representation that allows us to efficiently evaluate base certificates. A base certificate is any procedure that can provably guarantee that the prediction $f_n$ for a specific node $n$ cannot be changed by any perturbed graph in an admissible set, such as sparsity-aware smoothing (Bojchevski et al., 2020). As you shall see in the next section, our collective certificate requires evaluating base certificates for varying adversarial budgets within $\mathbb{L} = [r_{\boldsymbol{X}_{\mathrm{add}}}] \times [r_{\boldsymbol{X}_{\mathrm{del}}}] \times [r_{\boldsymbol{A}_{\mathrm{add}}}] \times [r_{\boldsymbol{A}_{\mathrm{del}}}]$ (with $[k] = \{0, \dots, k\}$), the set of vectors that do not exceed the collective global budget.

A base certificate implicitly partitions $\mathbb{L}$ into a set of budgets $\mathbb{K}^{(n)} \subseteq \mathbb{L}$ for which the prediction $f_n$ is certifiably robust and its complement $\overline{\mathbb{K}^{(n)}} = \mathbb{L} \setminus \mathbb{K}^{(n)}$ with

$$\mathbb{K}^{(n)} \subseteq \left\{ \boldsymbol{\rho} \in \mathbb{N}_0^4 \mid \boldsymbol{\rho} \in \mathbb{L} \wedge \forall (\boldsymbol{X}', \boldsymbol{A}') \in \mathbb{B}_{\mathcal{G}}(\boldsymbol{\rho}) : f_n(\boldsymbol{X}, \boldsymbol{A}) = f_n(\boldsymbol{X}', \boldsymbol{A}') \right\}. \tag{3}$$

---

[3]Note that the adversary only needs to control one of the nodes incident to an edge in order to perturb it. We do not associate different costs for perturbing different nodes or edges, but such an extension is straightforward.

Note the subset relation. The set $\mathbb{K}^{(n)}$ does not have to contain all budgets for which $f_n$ is robust. Conversely, a certain budget vector $\boldsymbol{\rho}$ not being part of $\mathbb{K}^{(n)}$ does not necessarily mean that $f_n$ can be attacked under threat model $\mathbb{B}_{\mathcal{G}}(\boldsymbol{\rho})$ – its robustness is merely unknown. We now make the following natural assumption about base certificates: If a classifier is certifiably robust to perturbations with a large global budget, it should also be certifiably robust to perturbations with a smaller global budget.

$$\forall \boldsymbol{\rho} \in \mathbb{K}^{(n)}, \boldsymbol{\rho}' \in \mathbb{L} : [\forall d \in \{1,2,3,4\} : \rho_d' \leq \rho_d] \implies \left[ \boldsymbol{\rho}' \in \mathbb{K}^{(n)} \right]. \tag{4}$$

From a geometric point of view, Eq. 4 means that the budgets $\mathbb{K}$ for which the prediction $f_n$ is certifiably robust form a singular enclosed volume around $\begin{pmatrix} 0 & 0 & 0 & 0 \end{pmatrix}^T$ within the larger volume $\mathbb{L}$. Determining whether a classifier is robust to perturbations in $\mathbb{B}_{\mathcal{G}}(\boldsymbol{\rho})$ is equivalent to determining which side of the surface enclosing the volume $\mathbb{K}$ the budget vector $\boldsymbol{\rho}$ lies on. This can be be done by evaluating linear inequalities, as shown in the following.

First, let us assume that all but one of the budgets are zero, e.g. $\mathbb{L} = [r_{\boldsymbol{X}_{\mathrm{add}}}] \times [0] \times [0] \times [0]$, with $r_{\boldsymbol{X}_{\mathrm{add}}} > 0$. Due to Eq. 4 there must be a distinct value $p_n \in \mathbb{N}_0$ (the smallest uncertifiable budget) with $\forall \boldsymbol{\rho} \in \mathbb{L} : \boldsymbol{\rho} \in \overline{\mathbb{K}^{(n)}} \iff \rho_1 \geq p_n$. Evaluating the base certificate can thus be performed by evaluating a single inequality. This approach can be generalized to arbitrary types of perturbations. Instead of using a single scalar $p_n$, we characterize the volume of budgets $\mathbb{K}^{(n)}$ via the pareto front of points on its enclosing surface:

$$\mathbb{P}^{(n)} = \left\{ \boldsymbol{\rho} \in \overline{\mathbb{K}^{(n)}} \mid \neg \exists \boldsymbol{\rho}' \in \overline{\mathbb{K}^{(n)}} : \boldsymbol{\rho}' \neq \boldsymbol{\rho} \wedge \forall d \in \{1,2,3,4\} : \rho_d' \leq \rho_d \right\}. \tag{5}$$

These points fulfill $\forall \boldsymbol{\rho} \in \mathbb{L} \left( \boldsymbol{\rho} \in \overline{\mathbb{K}^{(n)}} \iff \exists \boldsymbol{p} \in \mathbb{P}^{(n)}, \forall d \in \{1,2,3,4\} : \rho_d \geq p_d \right)$. Here, evaluating the base certificate can be performed by evaluating $4|\mathbb{P}|$ inequalities.

In the following, we assume that we are directly given this pareto front (or the smallest uncertifiable budget). Finding the pareto front can be easily implemented via a flood-fill algorithm that identifies the surface of volume $\mathbb{K}^{(n)}$, followed by a thinning operation (for more details, see Section D).

## 4 COLLECTIVE CERTIFICATE

To improve clarity in this section, we only discuss the global budget constraints. All remaining constraints from the threat model can be easily modelled as linear constraints. You can find the certificate for the full threat model in Section C. We first formalize the naïve collective certificate described in the introduction, which implicitly allows the adversary to use different graphs to attack different predictions. We then derive the proposed collective certificate, first focusing on attribute additions before extending it to arbitrary perturbations. We relax the certificate to a linear program to enable fast computation and show the certificate's tightness when using a randomized smoothing base certificate. We conclude by discussing the certificate's time complexity and limitations.

**Naïve collective certificate.** Assume we are given a clean input $(\boldsymbol{X}, \boldsymbol{A})$, a multi-output classifier $f$, a set $\mathbb{T}$ of target nodes and a set of admissible perturbed graphs $\mathbb{B}_{\mathcal{G}}$ fulfilling collective global budget constraints given by $r_{\boldsymbol{X}_{\mathrm{add}}}, r_{\boldsymbol{X}_{\mathrm{del}}}, r_{\boldsymbol{A}_{\mathrm{add}}}, r_{\boldsymbol{A}_{\mathrm{del}}}$. Let $\mathbb{L} = [r_{\boldsymbol{X}_{\mathrm{add}}}] \times [r_{\boldsymbol{X}_{\mathrm{del}}}] \times [r_{\boldsymbol{A}_{\mathrm{add}}}] \times [r_{\boldsymbol{A}_{\mathrm{del}}}]$ be the set of all vectors that that do not exceed the collective global budget. Further assume that the base certificate guarantees that each classifier $f_n$ is certifiable robust to perturbations within a set of budgets $\mathbb{K}^{(n)}$ (see Eq. 3). As discussed, the naïve certificate simply counts the predictions whose robustness to perturbations from $\mathbb{B}_{\mathcal{G}}$ is guaranteed by the base certificate. Using the representation of base certificates introduced in Section 3, this can be expressed as $\sum_{n \in \mathbb{T}} \mathbf{1} \left[ \mathbb{K}^{(n)} = \mathbb{L} \right]$. From the definition of $\mathbb{K}^{(n)}$ in Eq. 3, we can directly see that this is a lower bound on the optimal value of $\sum_{n \in \mathbb{T}} \min_{(\boldsymbol{X}', \boldsymbol{A}') \in \mathbb{B}_{\mathcal{G}}} \mathbf{1} \left[ f_n(\boldsymbol{X}, \boldsymbol{A}) = f_n(\boldsymbol{X}', \boldsymbol{A}') \right]$, i.e. the number of predictions guaranteed to be stable under attack. Note that each summand involves a different minimization problem, meaning the adversary may use a different graph to attack each of the nodes.

**Collective certificate for attribute additions.** To improve upon the naïve certificate, we want to determine the number of predictions that are simultaneously robust to attacks with a *single* graph:

$$\min_{(\boldsymbol{X}', \boldsymbol{A}') \in \mathbb{B}_{\mathcal{G}}} \sum_{n \in \mathbb{T}} \mathbf{1} \left[ f_n(\boldsymbol{X}, \boldsymbol{A}) = f_n(\boldsymbol{X}', \boldsymbol{A}') \right]. \tag{6}$$

Solving this problem is usually not tractable. For simplicity, let us assume that the adversary is only allowed to perform attribute additions. As before, we can lower-bound the indicator functions using the base certificates:

$$\min_{(\boldsymbol{X}',\boldsymbol{A}')\in\mathbb{B}_\mathcal{G}} \sum_{n\in\mathbb{T}} \mathbf{1}\left[\begin{pmatrix} b & 0 & 0 & 0 \end{pmatrix}^T \in \mathbb{K}^{(n)}\right] \tag{7}$$

where $b = \sum_{(n,d):X_{n,d}=0} X'_{n,d}$ is the number of attribute additions for a given perturbed graph. Since this certificate only depends on the number of perturbations, it is sufficient to optimize over the number of attribute additions while enforcing the global budget constraint:

$$\min_{b\in\mathbb{N}_0} \sum_{n\in\mathbb{T}} \mathbf{1}\left[\begin{pmatrix} b & 0 & 0 & 0 \end{pmatrix}^T \in \mathbb{K}^{(n)}\right] \text{ s.t. } b \le r_{\boldsymbol{X}_{\mathrm{add}}}. \tag{8}$$

There are two limitations: (1) The certificate does not account for locality, but simply considers the number of perturbations in the entire graph. In this regard, it is no different from the naïve collective certificate; (2) Evaluating the indicator functions, i.e. certifying the individual nodes, might involve complex optimization problems that are difficult to optimize through. We tackle (1) by evaluating the base certificates locally.

**Lemma 1** *Assume multi-output classifier $f$, corresponding receptive field indicators $\boldsymbol{\psi}^{(n)} \in \{0,1\}^N$ and $\boldsymbol{\Psi}^{(n)} \in \{0,1\}^{N\times N}$, and a clean graph $(\boldsymbol{X},\boldsymbol{A})$. Let $\mathbb{K}^{(n)}$ be the set of certifiable global budgets of prediction $f_n$, as defined in Eq. 3. Let $(\boldsymbol{X}',\boldsymbol{A}')$ be a perturbed graph. Define $\boldsymbol{X}'' \in \{0,1\}^{N\times D}$ and $\boldsymbol{A}'' \in \{0,1\}^{N\times N}$ as follows:*

$$\boldsymbol{X}''_{i,d} = \psi_i^{(n)} \boldsymbol{X}'_{i,d} + (1-\psi_i^{(n)})\boldsymbol{X}_{i,d}, \tag{9}$$

$$\boldsymbol{A}''_{i,j} = \Psi_{i,j}^{(n)} \boldsymbol{A}'_{i,j} + (1-\Psi_{i,j}^{(n)})\boldsymbol{A}_{i,j}, \tag{10}$$

*i.e. use values from the clean graph for bits that are not in $f_n$'s receptive field. If there exists a vector of budgets $\boldsymbol{\rho} \in \mathbb{N}_0^4$ such that $(\boldsymbol{X}'',\boldsymbol{A}'') \in \mathbb{B}_\mathcal{G}(\boldsymbol{\rho})$ and $\boldsymbol{\rho} \in \mathbb{K}^{(n)}$, then $f_n(\boldsymbol{X},\boldsymbol{A}) = f_n(\boldsymbol{X}',\boldsymbol{A}')$.*

See proof in Section B. Due to Lemma 1 we can ignore all perturbations outside $f_n$'s receptive field when evaluating its base certificate. We can thus replace $\begin{pmatrix} b & 0 & 0 & 0 \end{pmatrix}^T$ in Eq. 8 with $\begin{pmatrix} \boldsymbol{b}^T\boldsymbol{\psi}^{(n)} & 0 & 0 & 0 \end{pmatrix}^T$, where the vector $\boldsymbol{b} \in \mathbb{N}_0^N$ indicates the number of attribute additions at each node. Optimizing over $\boldsymbol{b}$ yields a collective certificate that accounts for locality:

$$\min_{\boldsymbol{b}\in\mathbb{N}_0} \sum_{n\in\mathbb{T}} \mathbf{1}\left[\begin{pmatrix} \boldsymbol{b}^T\boldsymbol{\psi}^{(n)} & 0 & 0 & 0 \end{pmatrix}^T \in \mathbb{K}^{(n)}\right] \text{ s.t. } ||\boldsymbol{b}||_1 \le r_{\boldsymbol{X}_{\mathrm{add}}}. \tag{11}$$

We now tackle issue (2) by employing the compact representation of base certificates defined in Section 3. Since we are only allowing one type of perturbation, the base certificate of each classifier $f_n$ is characterized by the smallest uncertifiable radius $p_n$ (see Section 3). To evaluate the indicator function in Eq. 11 we simply have to compare the number of perturbations in $f_n$'s receptive field to $p_n$, which can be implemented via the following MILP:

$$\min_{\boldsymbol{b}\in\mathbb{N}_0^N, \boldsymbol{t}\in\{0,1\}^N} |\mathbb{T}| - \sum_{n\in\mathbb{T}} t_n \tag{12}$$

$$\text{s.t. } \boldsymbol{b}^T\boldsymbol{\psi}^{(n)} \ge p_n t_n \quad \forall n \in \{1,\dots,N\} \tag{13}$$

$$||\boldsymbol{b}||_1 \le r_{\boldsymbol{X}_{\mathrm{add}}}. \tag{14}$$

Eq. 14 ensures that the number of perturbations fulfills the global budget constraint. Eq. 13 ensures that the indicator $t_n$ can only be set to 1 if the local perturbation on the l.h.s. exceeds or matches $p_n$, i.e. $f_n$ is *not* robustly certified by the base certificate. The adversary tries to minimize the number of robustly certified predictions in $\mathbb{T}$ (see Eq. 8), which is equivalent to Eq. 12.

**Collective certificate for arbitrary perturbations**. Lemma 1 holds for arbitrary perturbations. We only have to consider the perturbations within a prediction's receptive field when evaluating its base certificate. However, when multiple perturbation types are allowed, the base certificate of a prediction $f_n$ is not characterized by a scalar $p_n$, but by its pareto front $\mathbb{P}^{(n)}$ (see Eq. 5). Let $\boldsymbol{P}^{(n)} \in \mathbb{N}_0^{|\mathbb{P}^{(n)}|\times 4}$ be a matrix encoding of the set $\mathbb{P}^{(n)}$. To determine if $f_n$ is robust we can check whether

there is some pareto-optimal point $\boldsymbol{P}_{i,:}^{(n)}$ such that the amount of perturbation in $f_n$'s receptive field matches or exceeds $\boldsymbol{P}_{i,:}^{(n)}$ in all four dimensions. This can again be expressed as a MILP (see Eq. 15 to Eq. 23 below).

As before, we use a vector $\boldsymbol{t}$ with $t_n = 1$ indicating that $f_n$ is not certified by the base certificate. The adversary tries to find a budget allocation (parameterized by $\boldsymbol{b}_{\boldsymbol{X}_{\mathrm{add}}}, \boldsymbol{b}_{\boldsymbol{X}_{\mathrm{del}}}$ and $\boldsymbol{B}_{\boldsymbol{A}}$) that minimizes the number of robustly certified predictions in $\mathbb{T}$ (see Eq. 15). Eq. 20 and Eq. 21 ensure that the budget allocation is consistent with the global budget parameters characterizing $\mathbb{B}_{\mathcal{G}}$. The value of $t_n$ is determined by the following constraints: First, Eq. 17 to Eq. 19 ensure that $Q_{p,d}^{(n)}$ is only set to 1 if the local perturbation matches or exceeds the pareto-optimal point corresponding to row $p$ of $\boldsymbol{P}^{(n)}$ in dimension $d$. The constraints in Eq. 16 implement logic operations on $\boldsymbol{Q}^{(n)}$: Indicator $s_p^{(n)}$ can only be set to 1 if $\forall d \in \{1,2,3,4\} : Q_{p,d}^{(n)} = 1$. Indicator $t_n$ can only be set to 1 if $\exists p \in \{1, \ldots, |\mathbb{P}^{(n)}|\} : s_p^{(n)} = 1$. Combined, these constraints enforce that if $t_n = 1$, there must be some point in $\mathbb{P}^{(n)}$ that is exceeded or matched by the amount of perturbation in all four dimensions.

$$\min_{\left(\boldsymbol{Q}^{(n)}, \boldsymbol{s}^{(n)}\right)_{n=1}^{N}, \boldsymbol{b}_{\boldsymbol{X}_{\mathrm{add}}}, \boldsymbol{b}_{\boldsymbol{X}_{\mathrm{del}}}, \boldsymbol{B}_{\boldsymbol{A}}, \boldsymbol{t}} |\mathbb{T}| - \sum_{n \in \mathbb{T}} t_n \tag{15}$$

$$\text{s.t.} \quad ||\boldsymbol{s}^{(n)}||_1 \geq t_n, \qquad Q_{p,d}^{(n)} \geq s_p^{(n)}, \tag{16}$$

$$(\boldsymbol{b}_{\boldsymbol{X}_{\mathrm{add}}})^T \boldsymbol{\psi}^{(n)} \geq Q_{i,1}^{(n)} P_{p,1}^{(n)}, \qquad (\boldsymbol{b}_{\boldsymbol{X}_{\mathrm{del}}})^T \boldsymbol{\psi}^{(n)} \geq Q_{p,2}^{(n)} P_{p,2}^{(n)}, \tag{17}$$

$$\sum_{m,m' \leq N} (1 - A_{m,m'})(\boldsymbol{\Psi}^{(n)} \odot B_{\boldsymbol{A}})_{m,m'} \geq Q_{p,3}^{(n)} P_{p,3}^{(n)}, \tag{18}$$

$$\sum_{m,m' \leq N} A_{m,m'}(\boldsymbol{\Psi}^{(n)} \odot B_{\boldsymbol{A}})_{m,m'} \geq Q_{p,4}^{(n)} P_{p,4}^{(n)}, \tag{19}$$

$$||b_{\boldsymbol{X}_{\mathrm{add}}}||_1 \leq r_{\boldsymbol{X}_{\mathrm{add}}}, \qquad ||b_{\boldsymbol{X}_{\mathrm{add}}}||_1 \leq r_{\boldsymbol{X}_{\mathrm{del}}}, \tag{20}$$

$$\sum_{(i,j): A_{i,j}=0} B_{\boldsymbol{A}i,j} \leq r_{\boldsymbol{A}_{\mathrm{add}}}, \qquad \sum_{(i,j): A_{i,j}=1} B_{\boldsymbol{A}i,j} \leq r_{\boldsymbol{A}_{\mathrm{del}}}, \tag{21}$$

$$\boldsymbol{s}^{(n)} \in \{0,1\}^{|\mathbb{P}^{(n)}|}, \qquad \boldsymbol{Q}^{(n)} \in \{0,1\}^{|\mathbb{P}^{(n)}| \times 4} \qquad \boldsymbol{t} \in \{0,1\}^N \tag{22}$$

$$\boldsymbol{b}_{\boldsymbol{X}_{\mathrm{add}}}, \boldsymbol{b}_{\boldsymbol{X}_{\mathrm{del}}} \in \mathbb{N}_0^N, \qquad \boldsymbol{B}_{\boldsymbol{A}} \in \{0,1\}^{N \times N}. \tag{23}$$

**LP-relaxation.** For large graphs, finding an optimum to the mixed-integer problem is prohibitively expensive. In practice, we relax integer variables to reals and binary variables to $[0,1]$. Semantically, the relaxation means that bits can be partially perturbed, nodes can be partially controlled by the attacker and classifiers can be partially uncertified (i.e. $1 > t_n > 0$). The relaxation yields a linear program, which can be solved much faster.

**Tightness for randomized smoothing.** One recent method for robustness certification is randomized smoothing (Cohen et al., 2019). In randomized smoothing, a (potentially non-deterministic) base classifier $h : \mathbb{X} \mapsto \mathbb{Y}$ that maps from some input space $\mathbb{X}$ to a set of labels $\mathbb{Y}$ is transformed into a smoothed classifier $g(x)$ with $g(x) = \mathrm{argmax}_{y \in \mathbb{Y}} \Pr[h(\phi(x) = y]$, where $\phi(x)$ is some randomization scheme parameterized by input $x$. For the smoothed $g(x)$ we can then derive probabilistic robustness certificates. Randomized smoothing is a black-box method that only depends on $h$'s expected output behavior under $\phi(x)$ and does not require any further assumptions. Building on prior randomized smoothing work for discrete data by Lee et al. (2019), Bojchevski et al. (2020) propose a smoothing distribution and corresponding certificate for graphs. Using their method as a base certificate to our collective certificate, the resulting (non-relaxed) certificate is tight. That is, our mixed-integer collective certificate is the best certificate we can obtain for the specified threat model, if we do not use any information other than the classifier's expected predictions and their locality. Detailed explanation and proof in Section E.

**Time complexity.** For our method we need to construct the pareto fronts corresponding to each prediction's base certificate. This has to be performed only once and the results can then be reused in evaluating the collective certificate with varying parameters. We discuss the details of this preprocessing in Section D. The complexity of the collective certificate is based on the number of constraints and variables of the underlying (MI)LP. In total, we have $13 \sum_{n=1}^{N} |\mathbb{P}^{(n)}| + 8N + 2e + 5$

constraints and $5 \sum_{n=1}^{N} |\mathbb{P}^{(n)}| + 4N + e$ variables, where $e$ are the number of edges in the unperturbed graph (we disallow edge additions). For single-type perturbations we have $\mathcal{O}(N + e)$ terms, linear in the number of nodes and edges. The relaxed LP takes at most a few seconds to certify robustness for single-type perturbations and a few minutes for multiple types of perturbations (see Section 5).

**Limitations.** The proposed approach is designed to exploit locality. Without locality, it is equivalent to a naïve combination of base certificates that sums over perturbations in the entire graph. A non-obvious limitation is that our notion of locality breaks down if the receptive fields are data-dependent and can be arbitrarily extended by the adversary. Recall how we specified locality in Eq. 2: The indicators $\psi^{(n)}$ and $\Psi^{(n)}$ correspond to the union of all achievable receptive fields. Take for example a two-layer message-passing neural networks and an adversary that can add new edges. Each node is classified based on its 2-hop neighborhood. For any two nodes $n, m$, the adversary can construct a graph such that $m$ is in $f_n$ receptive field. We thus have to treat the $f_n$ as global, even if for any single graph they might only process some subgraph. Nonetheless, our method still yields significant improvements for edge deletions and arbitrary attribute perturbations. As discussed in prior work (Zügner & Günnemann, 2020) edge addition is inherently harder and less relevant in practice.

## 5 EXPERIMENTAL EVALUATION

**Experimental setup.** We evaluate the proposed approach by certifying node classifiers on multiple graphs and with different base certificates. We use 20 nodes per class to construct a train and a validation set. We certify all remaining nodes. We repeat each experiment five times with different random initializations and data splits. Unless otherwise specified, we do not impose any local budget constraints or constraints on the number of attacker-controlled nodes. We compare the proposed method with the naïve collective certificate, which simply counts the number of predictions that are certified to be robust by the base certificate. All experiments are based on the relaxed linear programming version of the certificate. We assess the integrality gap to the mixed-integer version in Section A. The code is publicly available under https://www.daml.in.tum.de/collective-robustness/. We also uploaded the implementation as supplementary material.

**Datasets, models and base certificates.** We train and certify models on the following datasets: Cora-ML (McCallum et al. (2000); Bojchevski & Günnemann (2018); $N = 2810$, 7981 edges, 7 classes), Citeseer (Sen et al. (2008); $N = 2110$, 3668 edges, 6 classes), PubMed (Namata et al. (2012); $N = 19717$, 44324 edges, 3 classes), Reuters-21578 [4] ($N = 862$, 2586 edges, 4 classes) and WebKB (Craven et al. (1998); $N = 877$, 2631 edges, 5 classes). The graphs for the natural language corpora Reuters and WebKB are constructed using the procedure described in Zhou & Vorobeychik (2020), resulting in 3-regular graphs. We use five types of classifiers: Graph convolution networks (GCN) (Kipf & Welling, 2017), graph attention networks (GAT) (Veličković et al., 2018), APPNP (Gasteiger et al., 2019), robust graph convolution networks (RGCN) (Zhu et al., 2019) and soft medoid aggregation networks (SMA) (Geisler et al., 2020). All classifiers are configured to have two layers, i.e. each node's classifier is dependent on its two-hop neighborhood. We use two types of base certificates: (Bojchevski et al., 2020) (randomized smoothing, arbitrary perturbations) and Zügner & Günnemann (2019) (convex relaxations of network nonlinearities, attribute perturbations). We provide a summary of all hyperparameters in Section F.

**Evaluation metrics.** We report the *certified ratio* on the test set, i.e. the percentage of nodes that are certifiably robust under a given threat model, averaged over all data splits. We further calculate the standard sample deviation in certified ratio (visualized as shaded areas in plots) and the average wall-clock time per collective certificate. For experiments in which only one global budget parameter is altered, we report the *average certifiable radius*, i.e. $\bar{r} = (\sum_{r=0}^{\infty} \omega(r) * r / \sum_{r=0}^{\infty} \omega(r))$, where $\omega(r)$ is the certified ratio for value $r$ of the global budget parameter, averaged over all splits.

**Attribute perturbations.** We first evaluate the certificate for a single perturbation type. Using randomized smoothing as the base certificate, we evaluate the certified ratio of GCN classifiers for varying global attribute deletion budgets $r_{\boldsymbol{X}_{\text{del}}}$ on the citation graphs Cora, Citeseer and PubMed (for Reuters and WebKB, see Section A). The remaining global budget parameters are set to $0$. Fig. 2 shows that for all datasets, the proposed method yields significantly larger certified ratios than the naïve certificate and can certify robustness for much larger $r_{\boldsymbol{X}_{\text{del}}}$. The average certifiable radius $\bar{r}$

---

[4]Distribution 1.0, available from http://www.daviddlewis.com/resources/testcollections/reuters21578

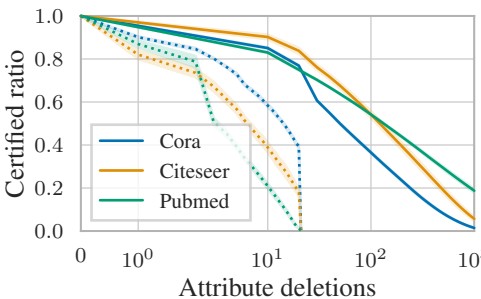
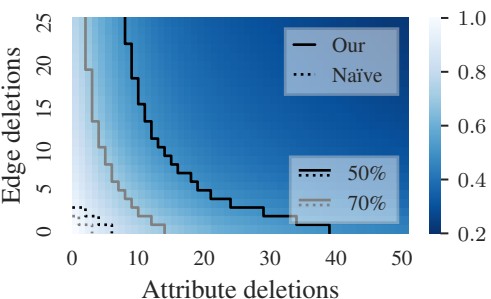

Figure 2: Certified ratios for smoothed GCN on Cora, Citeseer and PubMed, under varying $r_{\boldsymbol{X}_{\mathrm{del}}}$. We compare the proposed certificate (solid lines) to the naïve certificate (dotted lines). Our method certifies orders of magnitude larger radii (note the logarithmic x-axis).

Figure 3: Two-dimensional colective certificate for smoothed GCN on Cora-ML under varying $r_{\boldsymbol{X}_{\mathrm{del}}}$ and $r_{\boldsymbol{A}_{\mathrm{del}}}$. The solid and dotted contour lines show ratios $\geq 0.5$ and $\geq 0.7$ for our vs. the naïve certificate respectively. Our method achieves much larger certified ratios and radii.

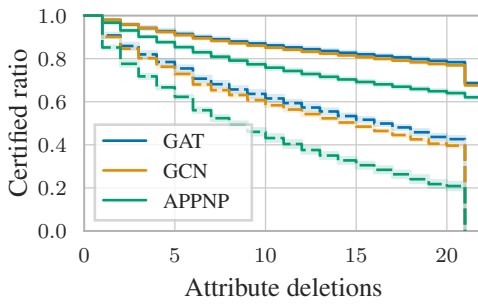
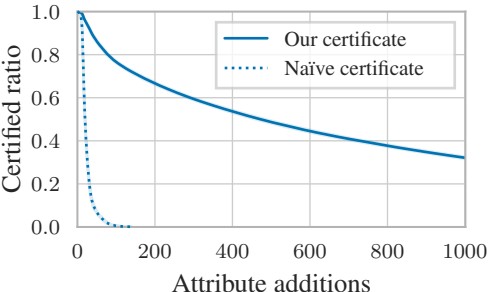

Figure 4: Comparison of certified ratios for GAT, GCN and APPNP on Cora-ML under varying $r_{\boldsymbol{X}_{\mathrm{del}}}$ for our (solid lines) and the naïve (dotted lines) collective certificate.

Figure 5: Certifying GCN on Citeseer, under varying $r_{\boldsymbol{X}_{\mathrm{add}}}$ using Zügner & Günnemann (2019)'s base certificate. Our certificate yields significantly larger certified ratios and radii.

on Citeseer increases from $7.18$ to $351.73$. The results demonstrate the benefit of using the collective certificate, which explicitly models simultaneous attacks on all predictions. The average wall-clock time per certificate on Cora, Citeseer and PubMed is $2.0\,\mathrm{s}$, $0.29\,\mathrm{s}$ and $336.41\,\mathrm{s}$. Interestingly, the base certificate yields the highest certifiable ratios on Cora, while the collective certificate yields the highest certifiable ratios on PubMed. We attribute this to differences in graph structure, which are explicitly taken into account by the proposed certification procedure.

**Simultaneous attribute and graph perturbations.** To evaluate the multi-dimensional version of the certificate, we visualize the certified ratio of randomly smoothed GCN classifiers for different combinations of $r_{\boldsymbol{X}_{\mathrm{del}}}$ and $r_{\boldsymbol{A}_{\mathrm{del}}}$ on Cora-ML. For an additional experiment on simultaneous attribute additions and deletions, see Section A. Fig. 3 shows that we achieve high certified ratios even when the attacker is allowed to perturb both the attributes and the structure. Comparing the contour lines at $50\,\%$ the naïve certiface can only certify much smaller radii, e.g. at most 6 attribute deletions compared to 39 for our approach. The average wall-clock time per certificate is $106.90\,\mathrm{s}$.

**Different classifiers.** Our method is agnostic towards classifier architectures, as long as they are compatible with the base certificate and their receptive fields can be determined. In Fig. 4 we compare the certified collective robustness of GAT, GCN, and APPNP, using the sparse smoothing certificate on Cora-ML.[5] Better base certificates translate into better collective certificates. For an additional experiment on the benefits of robust classifier architectures RGCN and SMA, see Section A.

---

[5]Our method can certify larger radii, $r \geq 10^3$ (see Fig. 2). Here we show $r \leq 20$ to highlight the difference.

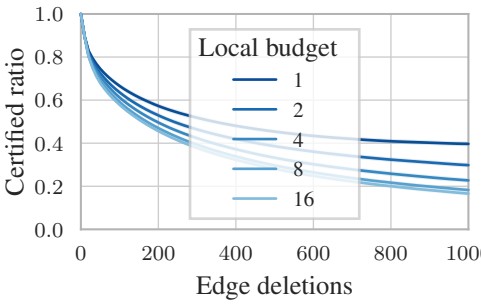 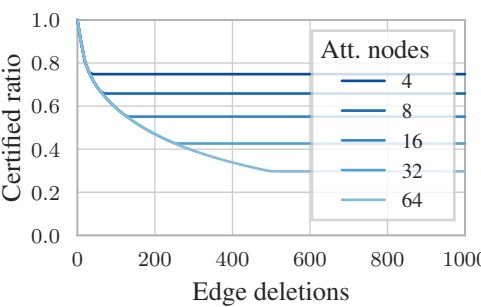

Figure 6: Certified ratios for smoothed GCN on Cora-ML, under varying $r_{A_{\mathrm{del}}}$ and $r_{A_{\mathrm{del,loc}}}$. Stricter local budgets yield larger certified ratios.

Figure 7: Certified ratios for smoothed GCN on Cora-ML. We vary $r_{X_{\mathrm{del}}}$ and $\sigma$. The certified ratios remain constant and non-zero for large $r_{A_{\mathrm{del}}}$.

**Different base certificates.** Our method is also agnostic to the base certificate type. We show that it works equally well with base certificates other than randomized smoothing. Specifically, we use the method from Zügner & Günnemann (2019). We certify a GCN model for varying $r_{X_{\mathrm{add}}}$ on Cite-seer. Unlike randomized smoothing, this base certificate models local budget constraints. Using the default in the base certificate's reference implementation, we limit the number of attribute additions per node to $\lfloor 0.01D \rfloor = 21$ for both the base and the collective certificate. Fig. 5 shows that the proposed collective certificate is again significantly stronger. The average certified radius $\hat{r}$ increases from $17.12$ to $971.36$. The average wall-clock time per certificate is $0.39\,\mathrm{s}$.

**Local constraints.** We evaluate the effect of additional constraints in our threat model. We can enforce local budget constraints and limit the number of attacker-controlled nodes, even if they are not explicitly modeled by the base certificate. In Fig. 6, we use a smoothed GCN on Cora-ML and vary both the global budget for edge deletions, $r_{A_{\mathrm{del}}}$, and the local budgets $r_{A_{\mathrm{del,loc}}}$. Even though the base certificate does not support local budget constraints, reducing the number of admissible deletions per node increases the certified ratio as expected. For example, limiting the adversary to one deletion per node more than doubles the certified ratio at $r_{A_{\mathrm{del}}} = 1000$. In Fig. 7, we fix a relatively large local budget of 16 edge deletions per node (only $\sim 5\%$ of nodes on Cora-ML have a degree $> 16$) and vary the number of attacker-controlled nodes. We see that for any given number of attacker nodes, there is some point $r_{A_{\mathrm{del}}}$ after which the certified ratio curve becomes constant. This constant value is an an upper limit on the number of classifiers that can be attacked with a given local budget and number of attacker-controlled nodes. It is independent of the global budget.

## 6 Conclusion

We propose the first collective robustness certificate. Assuming predictions based on a single shared input, we leverage the fact that an adversary must use a single adversarial example to attack all predictions. We focus on Graph Neural Networks, whose locality guarantees that perturbations to the input graph only affect predictions in a close neighborhood. The proposed method combines many weak base certificates into a provably stronger collective certificate. It is agnostic towards network architectures and base certification procedures. We evaluate it on multiple semi-supervised node classification datasets with different classifier architectures and base certificates. Our empirical results show that the proposed collective approach yields much stronger certificates than existing methods, which assume that an adversary can attack predictions independently with different graphs.

## 7 Acknowledgements

This research was supported by the German Research Foundation, Emmy Noether grant GU 1409/2-1, the German Federal Ministry of Education and Research (BMBF), grant no. 01IS18036B, and the TUM International Graduate School of Science and Engineering (IGSSE), GSC 81.

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

## A ADDITIONAL EXPERIMENTS

**Robust architectures.** Our comparison of different standard classifier architectures demonstrated that the proposed collective certificate is architecture-agnostic and that better base certificates translate into better collective certificates. In Fig. 8 we assess the benefit of using SMA and RGCN, both of which are robust architectures meant to improve adversarial robustness. We use GCN as a baseline for comparison and evaluate the respective certified ratios for varying attribute deletion budgets on Cora-ML. While RGCN is supposed to be more robust to adversarial attacks, it has a lower certified ratio than GCN. Soft medoid aggregation on the other hand has a significantly higher certified ratio. Its base certificate is almost as strong as the collective certificate of RGCN. Its collective certified ratio at $r_{\boldsymbol{X}_{\mathrm{del}}} = 21$ is 88.3%, compared to the 76.9% and 74% of GCN and RGCN.

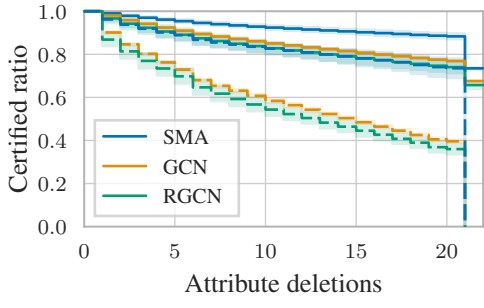
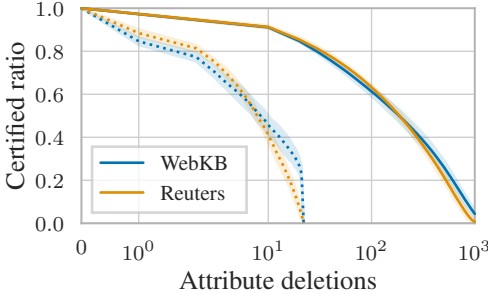

Figure 8: Certified ratios for smoothed GCN, RGCN and soft medoid aggregation on Cora-ML under varying $r_{\boldsymbol{X}_{\mathrm{del}}}$ for our (solid lines) and the naïve (dotted lines) collective certificate.

Figure 9: Certified ratios for smoothed GCN on WebKB and Reuters-21578, under varying $r_{\boldsymbol{X}_{\mathrm{del}}}$ for our (solid lines) and the naïve (dotted lines) collective certificate.

**Attribute perturbations on additional datasets.** In addition to citation graphs, we also use graphs constructed from the Reuters-21578 and WebKB natural language corpora to evaluate the proposed certificate. As in the main experiments section, we use randomized smoothing as a base certificate for GCN classifiers and assess the certified ratio for varying global attribute deletion budgets. Fig. 9 shows that the certified ratio increases and that much larger $r_{\boldsymbol{X}_{\mathrm{del}}}$ (up to approximately $10^3$) can be certified when using the collective approach. The average certifiable radius for Reuters and WebKB increases from 6.54 and 8.08 to 265.62 and 309.32, respectively. With less than 900 nodes each, both datasets are smaller than our three citation graphs. This leads to even shorter average wall-clock times per certificate: 0.105 s and 0.116 s.

**Simultaneous attribute deletions and additions.** In the main experiments section, we applied our collective certificate to simultaneous certification of attribute and adjacency deletions. Here, we assess how it performs for simultaneous deletions and additions of attributes. We again use a randomly smoothed GCN classifier on Cora-ML, and perform collective certification for different combinations of $r_{\boldsymbol{X}_{\mathrm{add}}}$ and $r_{\boldsymbol{X}_{\mathrm{del}}}$ on Cora-ML. As shown in Fig. 3, the collective certificate is again much stronger than the naïve collective certificate. For example, we obtain certified ratios between 30% and 60% at radii for which the naïve collective certificate cannot certify any robustness at all. The average wall-clock time per certificate is 40.51 s.

**Integrality gap.** For all previous experiments, we used the relaxed linear programming version of the certificate to reduce the compute time. To assess the integrality gap (i.e. the difference between the mixed-integer linear programming and the linear programming based certificates), we apply both versions of the certificate to a single smoothed GCN on Cora. We certify both robustness to attribute deletions (Fig. 11a) and edge deletions (Fig. 11b). The wall-clock time per certificate for the MILP increased from 0.24 s to 64 h with increasing edge deletion budget (0.35 s to 94 h for attribute deletions). Due to the exploding runtime for the MILP, we cannot compute the integrality gap for radii larger than 8 and 12, respectively.[6] The integrality gap is small (at most 4% for attribute deletions, 4.3% for edge deletions), relative to the certified ratio, and appears to be slightly increasing with increasing global budget.

---

[6]As discussed in the main paper the relaxed LP is fast and efficient to solve, even for radii larger than 1000.

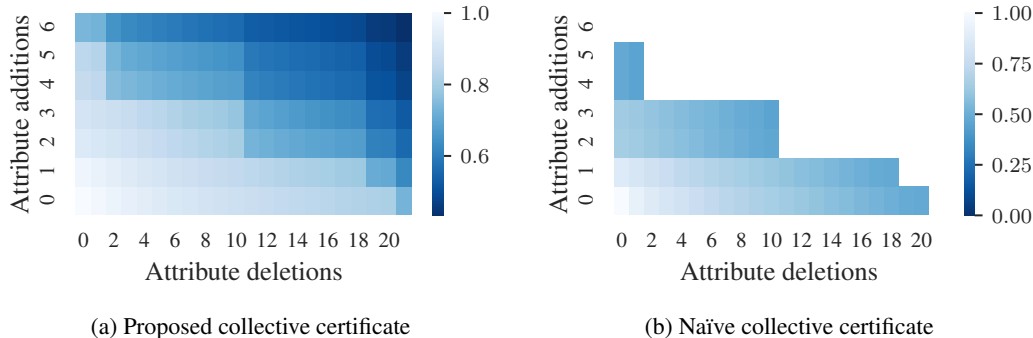

(a) Proposed collective certificate          (b) Naïve collective certificate

Figure 10: Comparison of the proposed collective certificate (Fig. 10a) to the naïve collective certificate (Fig. 10b) for certification of smoothed GCN on Cora-ML, under varying $r_{\mathbf{X}_{\mathrm{add}}}$ and $r_{\mathbf{X}_{\mathrm{del}}}$. Our method achieves much larger certified ratios for all combinations of attack radii.

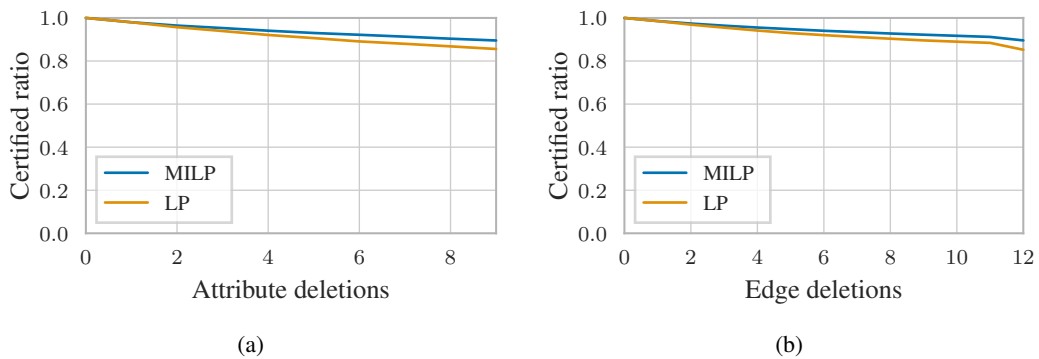

(a)          (b)

Figure 11: Certified ratios for smoothed GCN on Cora, under varying $r_{\mathbf{X}_{\mathrm{del}}}$ (Fig. 11a) and $r_{\mathbf{A}_{\mathrm{del}}}(Fig.\ 11b)$, using the mixed-integer collective certificate (blue line) and the relaxed linear programming certificate (orange line). The integrality gap is small, relative to the certified ratio.

## B    FORMAL DEFINITION OF THREAT MODEL PARAMETERS AND PROOFS

Here we define formally the set of admissible perturbed graphs $\mathbb{B}_{\mathcal{G}}$ described in Section 2. Recall that we have an unperturbed graph $(\mathbf{X}, \mathbf{A}) \in \mathbb{G}$, global budget parameters $r_{\mathbf{X}_{\mathrm{add}}}, r_{\mathbf{X}_{\mathrm{del}}}, r_{\mathbf{A}_{\mathrm{add}}}, r_{\mathbf{A}_{\mathrm{del}}} \in \mathbb{N}_0$, local budget parameters $\boldsymbol{r}_{\mathbf{X}_{\mathrm{add,loc}}}, \boldsymbol{r}_{\mathbf{X}_{\mathrm{del,loc}}}, \boldsymbol{r}_{\mathbf{A}_{\mathrm{add,loc}}}, \boldsymbol{r}_{\mathbf{A}_{\mathrm{del,loc}}} \in \mathbb{N}_0^N$ and at most $\sigma$ adversary-controlled nodes. Given these parameters, the set of admissible perturbed graphs $\mathbb{B}_{\mathcal{G}}$ is defined as follows:

$$(\mathbf{X}', \mathbf{A}') \in \mathbb{B}_{\mathcal{G}} \implies$$

$$\left|\{(n,d) : X_{n,d} = 0 \neq X'_{n,d}\}\right| \leq r_{\mathbf{X}_{\mathrm{add}}} \wedge \left|\{(n,d) : X_{n,d} = 1 \neq X'_{n,d}\}\right| \leq r_{\mathbf{X}_{\mathrm{del}}}$$

$$\wedge \left|\{(n,m) : A_{n,m} = 0 \neq A'_{n,m}\}\right| \leq r_{\mathbf{A}_{\mathrm{add}}} \wedge \left|\{(n,m) : A_{n,m} = 1 \neq A'_{n,m}\}\right| \leq r_{\mathbf{A}_{\mathrm{del}}}$$

$$\wedge \Big( \forall n \in \{1, \ldots, N\} :$$

$$\left|\{d : X_{n,d} = 0 \neq X'_{n,d}\}\right| \leq r_{\mathbf{X}_{\mathrm{add,loc}\, n}} \wedge \left|\{d : X_{n,d} = 1 \neq X'_{n,d}\}\right| \leq r_{\mathbf{X}_{\mathrm{del,loc}\, n}} \quad (24)$$

$$\wedge \left|\{m : A_{n,m} = 0 \neq X'_{n,m}\}\right| \leq r_{\mathbf{A}_{\mathrm{add,loc}\, n}} \wedge \left|\{m : A_{n,m} = 1 \neq X'_{n,m}\}\right| \leq r_{\mathbf{A}_{\mathrm{del,loc}\, n}} \Big)$$

$$\wedge \Big( \exists \mathbb{S} \subseteq \{1, \ldots, N\} : \Big( |\mathbb{S}| \leq \sigma \wedge \forall d \in \{1, \ldots, D\}, n, m \in \{1, \ldots, N\} :$$

$$\big( X'_{n,d} \neq X_{n,d} \implies n \in \mathbb{S} \big) \wedge \big( A'_{n,m} \neq A_{n,m} \implies n \in \mathbb{S} \vee m \in \mathbb{S} \big) \Big) \Big).$$

Next, we show the proof of Lemma 1 delegated from the main paper.

**Proof**: By the definition of receptive fields (Eq. 2) changes outside the receptive field do not influence the prediction, i.e. $f_n(\boldsymbol{X}', \boldsymbol{A}') = f_n(\boldsymbol{X}'', \boldsymbol{A}'')$. Since $(\boldsymbol{X}'', \boldsymbol{A}'') \in \mathbb{B}_{\mathcal{G}}(\boldsymbol{\rho})$ and $\boldsymbol{\rho} \in \mathbb{K}^{(n)}$, we know that $f_n(\boldsymbol{X}, \boldsymbol{A}) = f_n(\boldsymbol{X}'', \boldsymbol{A}'')$. By transitivity $f_n(\boldsymbol{X}, \boldsymbol{A}) = f_n(\boldsymbol{X}', \boldsymbol{A}') \square$.

## C  FULL COLLECTIVE CERTIFICATE

Here we discuss how to incorporate local budget constraints and constraints on the number of attacker-controlled nodes into the collective certificate for global budget constraints (see Eq. 15). We also discuss how to adapt it to undirected adjacency matrices (see Section C.1) As before, we lower-bound the true objective

$$\min_{(\boldsymbol{X}', \boldsymbol{A}') \in \mathbb{B}_{\mathcal{G}}} \sum_{n \in \mathbb{T}} \mathbf{1}\left[f_n(\boldsymbol{X}, \boldsymbol{A}) = f_n(\boldsymbol{X}', \boldsymbol{A}')\right]. \tag{25}$$

by replacing the indicator functions with evaluations of the corresponding base certificates and optimizing over the number of perturbations per node / the perturbed edges. The only difference is that more constraints are imposed on $\mathbb{B}_{\mathcal{G}}$.

The derivation proceeds as before: Lemma 1 still holds, meaning we only have to consider perturbations within $f_n$'s receptive field in evaluating whether its robustness is guaranteed by the base certificate. The base certificates can be efficiently evaluated by comparing the perturbation within $f_n$'s receptive field to all points in the pareto front $\mathbb{P}^{(n)}$ characterizing the volume of budgets $\mathbb{K}^{(n)}$. After encoding $\mathbb{P}^{(n)}$ as a matrix $\boldsymbol{P}^{(n)} \in \mathbb{N}_0^{|\mathbb{P}^{(n)}| \times 4}$, we can solve the following optimization problem to obtain a lower bound on Eq. 25:

$$\min_{\left(\boldsymbol{Q}^{(n)}, \boldsymbol{s}^{(n)}\right)_{n=1}^N, \boldsymbol{b}_{\boldsymbol{X}_{\text{add}}}, \boldsymbol{b}_{\boldsymbol{X}_{\text{del}}}, \boldsymbol{B}_{\boldsymbol{A}}, \boldsymbol{t}} |\mathbb{T}| - \sum_{n \in \mathbb{T}} t_n \tag{26}$$

$$\text{s.t.} \qquad ||\boldsymbol{s}^{(n)}||_1 \geq t_n, \qquad Q_{p,d}^{(n)} \geq s_p^{(n)}, \tag{27}$$

$$(\boldsymbol{b}_{\boldsymbol{X}_{\text{add}}})^T \boldsymbol{\psi}^{(n)} \geq Q_{i,1}^{(n)} P_{p,1}^{(n)}, \qquad (\boldsymbol{b}_{\boldsymbol{X}_{\text{del}}})^T \boldsymbol{\psi}^{(n)} \geq Q_{p,2}^{(n)} P_{p,2}^{(n)}, \tag{28}$$

$$\sum_{m,m' \leq N} (1 - A_{m,m'})(\boldsymbol{\Psi}^{(n)} \odot B_{\boldsymbol{A}})_{m,m'} \geq Q_{p,3}^{(n)} P_{p,3}^{(n)}, \tag{29}$$

$$\sum_{m,m' \leq N} A_{m,m'}(\boldsymbol{\Psi}^{(n)} \odot B_{\boldsymbol{A}})_{m,m'} \geq Q_{p,4}^{(n)} P_{p,4}^{(n)}, \tag{30}$$

$$||b_{\boldsymbol{X}_{\text{add}}}||_1 \leq r_{\boldsymbol{X}_{\text{add}}}, \qquad ||b_{\boldsymbol{X}_{\text{add}}}||_1 \leq r_{\boldsymbol{X}_{\text{del}}}, \tag{31}$$

$$\sum_{(i,j): A_{i,j}=0} B_{\boldsymbol{A}i,j} \leq r_{\boldsymbol{A}_{\text{add}}}, \qquad \sum_{(i,j): A_{i,j}=1} B_{\boldsymbol{A}i,j} \leq r_{\boldsymbol{A}_{\text{del}}}, \tag{32}$$

$$b_{\boldsymbol{X}_{\text{add}}\,n} \leq a_n r_{\boldsymbol{X}_{\text{add,loc}\,n}}, \qquad b_{\boldsymbol{X}_{\text{del}\,n}} \leq a_n r_{\boldsymbol{X}_{\text{del,loc}\,n}}, \tag{33}$$

$$\sum_{m:A_{n,m}=0} B_{\boldsymbol{A}n,m} + B_{\boldsymbol{A}m,n} \leq r_{\boldsymbol{A}_{\text{add,loc}\,n}}, \qquad \sum_{m:A_{n,m}=1} B_{\boldsymbol{A}n,m} + B_{\boldsymbol{A}m,n} \leq r_{\boldsymbol{A}_{\text{del,loc}\,n}} \tag{34}$$

$$B_{i,j} \leq a_i + a_j \; \forall i,j \in \{1,\ldots,N\}, \tag{35}$$

$$||\boldsymbol{a}||_1 \leq \sigma, \tag{36}$$

$$\boldsymbol{s}^{(n)} \in \{0,1\}^{|\mathbb{P}^{(n)}|}, \qquad \boldsymbol{Q}^{(n)} \in \{0,1\}^{|\mathbb{P}^{(n)}| \times 4}, \qquad \boldsymbol{t} \in \{0,1\}^N, \tag{37}$$

$$\boldsymbol{a} \in \{0,1\}^N, \qquad \boldsymbol{b}_{\boldsymbol{X}_{\text{add}}}, \boldsymbol{b}_{\boldsymbol{X}_{\text{del}}} \in \mathbb{N}_0^N, \qquad \boldsymbol{B}_{\boldsymbol{A}} \in \{0,1\}^{N \times N}, \tag{38}$$

$\forall n \in \{1,\ldots,N\}, p \in \{1,\ldots,|\mathbb{P}^{(n)}|\}, d \in \{1,\ldots,4\}$.

The constraints from Eq. 27 to Eq. 30 are identical to constraints Eq. 16 to Eq. 19 of our collective certificate for global budget constraints. They simply implement boolean logic to determine whether there is some pareto-optimal $\boldsymbol{p} \in \mathbb{P}(n)$ such that the perturbation in $f_n$'s receptive field matches or exceeds $\boldsymbol{p}$ in all four dimensions. If this is the case, the base certificate cannot certify the robustness of $f_n$ and $t_n$ can be set to 1. Eq. 31 and Eq. 32 enforce the global budget constraints. The difference to the global budget certificate lies in Eq. 33 to Eq. 36. We introduce an additional variable vector $\boldsymbol{a} \in \{0,1\}^N$ that indicates which nodes are attacker controlled. Eq. 33 enforces that the attributes of node

$n$ remain unperturbed, unless $a_n = 1$. If $a_n = 1$, the adversary can add or delete at most $r_{\boldsymbol{X}_{\text{del,loc}\,n}}$ or $r_{\boldsymbol{X}_{\text{del,loc}\,n}}$ attribute bits. With edge perturbations, it is sufficient for either incident node to be attacker-controlled. This is expressed via Eq. 35. The number of added or deleted edges incident to node $n$ is constrained via Eq. 34. Finally, Eq. 36 ensures that at most $\sigma$ nodes are attacker-controlled.

### C.1 Undirected adjacency matrix

To adapt our certificate to undirected graphs, we simply change the interpretation of the indicator matrix $\boldsymbol{B_A}$. Now, setting either $B_{\boldsymbol{A}i,j}$ or $B_{\boldsymbol{A}j,i}$ to 1 corresponds to perturbing the undirected edge $\{i,j\}$. An edge should not be perturbed twice, which we express through an additional constraint:

$$B_{\boldsymbol{A}i,j} + B_{\boldsymbol{A}j,i} \leq 1 \; \forall i,j \in \{1,\ldots,N\}. \tag{39}$$

We further combine Eq. 34, which enforced that at least one of the incident nodes of a perturbed edge has to be attacker controlled, and Eq. 35, which enforced the local budgets for edge perturbations, into following constraints:

$$\sum_{m:A_{n,m}=0} B_{\boldsymbol{A}n,m} \leq a_n r_{\boldsymbol{A}_{\text{add,loc}\,n}} \tag{40}$$

$$\sum_{m:A_{n,m}=1} B_{\boldsymbol{A}n,m} \leq a_n r_{\boldsymbol{A}_{\text{del,loc}\,n}} \tag{41}$$

These changes do not affect the optimal value of the mixed-integer linear program. But they are more effective than Eq. 34 and Eq. 35 when solving the relaxed linear program and the nodes' local budgets are small relative to their degree.

## D Determining the pareto front of base certificates

For our collective certificate, we assume that base certificates directly yield the pareto front $\mathbb{P}^{(n)}$ of points enclosing the volume of budgets $\mathbb{K}^{(n)}$ for which the prediction $f_n$ is certifiably robust:

$$\mathbb{P}^{(n)} = \left\{ \boldsymbol{\rho} \in \overline{\mathbb{K}^{(n)}} \mid \neg \exists \boldsymbol{\rho}' \in \overline{\mathbb{K}^{(n)}} : \boldsymbol{\rho}' \neq \boldsymbol{\rho} \wedge \forall d \in \{1,2,3,4\} : \rho_d' \leq \rho_d \right\} \tag{42}$$

with

$$\mathbb{K}^{(n)} \subseteq \left\{ \boldsymbol{\rho} \in \mathbb{N}_0^4 \mid \boldsymbol{\rho} \in \mathbb{L} \wedge \forall (\boldsymbol{X}', \boldsymbol{A}') \in \mathbb{B}_{\mathcal{G}}(\boldsymbol{\rho}) : f_n(\boldsymbol{X}, \boldsymbol{A}) = f_n(\boldsymbol{X}', \boldsymbol{A}') \right\}, \tag{43}$$

$\overline{\mathbb{K}^{(n)}} = \mathbb{L} \backslash \mathbb{K}^{(n)}$ and $\mathbb{L} = [r_{\boldsymbol{X}_{\text{add}}}] \times [r_{\boldsymbol{X}_{\text{del}}}] \times [r_{\boldsymbol{A}_{\text{add}}}] \times [r_{\boldsymbol{A}_{\text{del}}}]$ (with $[k] = \{0,\ldots,k\}$ (see Section 3). In practice, finding this representation requires some additional processing which we shall discuss in this section.

Existing certificates for graph-structured data are methods that determine for a specific budget $\boldsymbol{\rho} \in \mathbb{L}$ whether a classifier $f_n$ is robust to perturbations in $\mathbb{B}_{\mathcal{G}}(\boldsymbol{\rho})$. In other words: They can only test the membership relation $\boldsymbol{\rho} \in \mathbb{K}^{(n)}$. One possible way of finding the pareto front is through the following three-step process:

1. Use a flood-fill algorithm starting at $(0 \quad 0 \quad 0 \quad 0)^T$ to determine $\mathbb{K}^{(n)}$.
2. Identify all points in $\overline{\mathbb{K}^{(n)}} = \mathbb{L} \backslash \mathbb{K}^{(n)}$ that enclose the volume $\mathbb{K}^{(n)}$.
3. Remove all enclosing points that are not pareto-optimal.

A pseudo-code implementation is provided in algorithm 1. It has a running time in $\mathcal{O}\left(c \left| \mathbb{K}^{(n)} \right| \right)$, where $c$ is the worst-case case of performing a membership test $\boldsymbol{\rho} \in \mathbb{K}^{(n)}$.

---

**Algorithm 1:** Determining the pareto front of base certificates

---

**Result:** Set $\mathbb{P}^{(n)}$ of pareto-optimal points enclosing the volume of budgets $\mathbb{K}^{(n)}$ within
$\qquad \mathbb{L} = [r_{\boldsymbol{X}_{\mathrm{add}}}] \times [r_{\boldsymbol{X}_{\mathrm{del}}}] \times [r_{\boldsymbol{A}_{\mathrm{add}}}] \times [r_{\boldsymbol{A}_{\mathrm{del}}}]$ (see Section 3).
$\mathbb{P}'^{(n)} \leftarrow \{\}$ ; //Potential pareto points
closed_set $\leftarrow \{\}$ ; //Visited vectors
**Function** *flood_fill(ρ)*
 | closed_set $\leftarrow$ closed_set $\cup \{\boldsymbol{\rho}\}$ ;
 | **if** $\neg \left( \boldsymbol{\rho} \in \mathbb{K}^{(n)} \right)$ **then**
 |  | $\mathbb{P}'^{(n)} \leftarrow \mathbb{P}'^{(n)} \cup \{\boldsymbol{\rho}\}$ ;
 | **else**
 |  | **for** $\boldsymbol{\rho}' \in \{\boldsymbol{\rho} + \boldsymbol{d} | \boldsymbol{d} \in \{0,1\}^4 \wedge ||\boldsymbol{d}||_1 = 1\} \cap \mathbb{L}$ **do**
 |  |  | **if** $\neg (\boldsymbol{\rho}' \in closed\_set)$ **then**
 |  |  |  | *flood_fill(ρ′)* ; //Consider neighboring vectors
 |  |  | **end**
 |  | **end**
 | **end**
*flood_fill(*$(0 \quad 0 \quad 0 \quad 0)^T$*)* ;
$\mathbb{P}^{(n)} \leftarrow \{\}$ ; //Pareto front
**for** $\boldsymbol{\rho} \in \mathbb{P}^{(n)'}$ **do**
 | pareto_optimal $\leftarrow$ true ;
 | **for** $\boldsymbol{\rho}' \in \{\boldsymbol{\rho} - \boldsymbol{d}' | \boldsymbol{d}' \in \{0,1\}^4 \wedge ||\boldsymbol{d}'||_1 \geq 1\}$ **do**
 |  | **if** $\boldsymbol{\rho}' \in \mathbb{P}^{(n)'}$ **then**
 |  |  | pareto_optimal $\leftarrow$ false ; //Pareto-optimality does not allow
 |  |  |  decreasing values while staying in $\overline{\mathbb{K}^{(n)}}$
 |  |  | break;
 |  | **end**
 | **end**
 | **if** pareto_optimal **then**
 |  | $\mathbb{P}^{(n)} \leftarrow \mathbb{P}^{(n)} \cup \{\boldsymbol{\rho}\}$
 | **end**
**end**

---

## E  TIGHTNESS FOR RANDOMIZED SMOOTHING

In this section we prove that if we use the randomized smoothing based certificate from Bojchevski et al. (2020) as our base certificate, then our collective certificate is tight: If we do not make any further assumptions, outside each smoothed classifier's receptive field and its expected output behavior under the smoothing distribution, we cannot obtain a better collective certificate. We define the base certificate and then provide a constructive proof of the resulting collective certificate's tightness.

### E.1  RANDOMIZED SMOOTHING FOR SPARSE DATA

Bojchevski et al. (2020) provide a robustness certificate for classification of arbitrary sparse binary data. Applied to node classification, it can be summarized as follows:

Assume we are given a multi-output classifier $h : \mathbb{G} \mapsto \{1, \dots, C\}^N$. Define a smoothed classifier $f : \mathbb{G} \mapsto \{1, \dots, C\}^N$ with

$$f_n(\boldsymbol{X}, \boldsymbol{A}) = \mathrm{argmax}_{c \in \{1,\dots,C\}} \Pr\left[ h_n(\phi_{\mathrm{attr}}(\boldsymbol{X}), \phi_{\mathrm{adj}}(\boldsymbol{A})) = c \right] \quad \forall n \in \{1, \dots, N\}, \quad (44)$$

where $\phi_{\mathrm{attr}}$ and $\phi_{\mathrm{adj}}$ are two independent randomization schemes that assign probability mass to the set of attribute matrices $\{0,1\}^{N \times D}$ and adjacency matrices $\{0,1\}^{N \times N}$, respectively. The ran-

domization schemes are defined as follows:

$$\Pr\left[\phi_{\text{attr}}(\boldsymbol{X})_{m,d} = 1 - X_{m,d}\right] = \theta_{\boldsymbol{X}_{\text{del}}}^{X_{m,d}} \theta_{\boldsymbol{X}_{\text{add}}}^{(1-X_{m,d})} \quad \forall m \in \{1,\ldots,N\}, d \in \{1,\ldots,D\} \quad (45)$$

$$\Pr\left[\phi_{\text{adj}}(\boldsymbol{A})_{i,j} = 1 - A_{i,j}\right] = \theta_{\boldsymbol{A}_{\text{del}}}^{A_{i,j}} \theta_{\boldsymbol{A}_{\text{add}}}^{(1-A_{i,j})} \quad \forall i,j \in \{1,\ldots,N\}. \quad (46)$$

Each bit's probability of being flipped is dependent on its current value, but independent of the other bits.

An adversarially perturbed graph $(\boldsymbol{X}', \boldsymbol{A}')$ is successful in changing prediction $y_n = f_n(\boldsymbol{X}, \boldsymbol{A})$, if

$$y_n \neq \text{argmax}_{c \in \{1,\ldots,C\}} \Pr\left[h_n(\phi_{\text{attr}}(\boldsymbol{X}'), \phi_{\text{adj}}(\boldsymbol{A}')) = c\right]. \quad (47)$$

Evaluating this inequality is usually not tractable. We can however relax the problem: Let $p_n = \Pr\left[h_n(\phi_{\text{attr}}(\boldsymbol{X}), \phi_{\text{adj}}(\boldsymbol{A})) = y_n\right]$ and let $\mathbb{H}$ be the set of all possible classifiers for graphs in $\mathcal{G}$ (including non-deterministic ones). If

$$\left(\min_{\tilde{h}_n \in \mathbb{H}} \Pr\left[\tilde{h}_n(\phi_{\text{attr}}(\boldsymbol{X}'), \phi_{\text{adj}}(\boldsymbol{A}')) = y_n\right]\right) > 0.5 \quad (48)$$

$$\text{s.t.} \quad \Pr\left[\tilde{h}_n(\phi_{\text{attr}}(\boldsymbol{X}), \phi_{\text{adj}}(\boldsymbol{A})) = y_n\right] = p_n, \quad (49)$$

then $f_n(\boldsymbol{X}', \boldsymbol{A}') = f_n(\boldsymbol{X}, \boldsymbol{A})$. It is easy to see why: The unsmoothed $h_n$ is in the set defined by Eq. 49, so the result of the optimization problem is a lower bound on $\Pr\left[h_n(\phi_{\text{attr}}(\boldsymbol{X}'), \phi_{\text{adj}}(\boldsymbol{A}')) = y_n\right]$. If this lower bound is larger than 0.5, then $y_n$ is guaranteed to be the argmax class.

Optimizing over the set of all possible classifiers might appear hard. We can however use the approach of Lee et al. (2019) to find an optimum. Let $\boldsymbol{X}', \boldsymbol{A}'$ be a graph that results from $b_{\boldsymbol{X}_{\text{add}}}$ attribute additions, $b_{\boldsymbol{X}_{\text{del}}}$ attribute deletions, $b_{\boldsymbol{A}_{\text{add}}}$ edge additions, and $b_{\boldsymbol{A}_{\text{del}}}$ edge additions applied to $(\boldsymbol{X}, \boldsymbol{A})$. We can partition the set of all graphs $\mathbb{G}$ into $\left(b_{\boldsymbol{X}_{\text{add}}} + b_{\boldsymbol{X}_{\text{del}}} + 1\right)\left(b_{\boldsymbol{A}_{\text{add}}} + b_{\boldsymbol{A}_{\text{del}}} + 1\right)$ regions that have a constant likelihood ratio under our smoothing distribution:

$$\left\{\mathbb{J}_{q_{\boldsymbol{X}}, q_{\boldsymbol{A}}} \,\middle|\, q_{\boldsymbol{X}}, q_{\boldsymbol{A}} \in \mathbb{N}_0 \wedge q_{\boldsymbol{X}} \leq \left(b_{\boldsymbol{X}_{\text{add}}}^{(n)} + b_{\boldsymbol{X}_{\text{del}}}^{(n)} + 1\right) \wedge q_{\boldsymbol{A}} \leq \left(b_{\boldsymbol{A}_{\text{add}}}^{(n)} + b_{\boldsymbol{A}_{\text{del}}}^{(n)} + 1\right)\right\} \quad (50)$$

with

$$\left((\boldsymbol{X}'', \boldsymbol{A}'') \in \mathbb{J}_{q_{\boldsymbol{X}}, q_{\boldsymbol{A}}}\right) \implies \left(\frac{\Pr\left[\phi_{\text{attr}}(\boldsymbol{X}) = X'' \wedge \phi_{\text{adj}}(\boldsymbol{A}) = A''\right]}{\Pr\left[\phi_{\text{attr}}(\boldsymbol{X}') = X'' \wedge \phi_{\text{adj}}(\boldsymbol{A}') = A''\right]} = \eta_{q_{\boldsymbol{X}}, q_{\boldsymbol{A}}},\right) \quad (51)$$

where the $\eta_{\cdot, \cdot} \in \mathbb{R}_+$ are constants. The regions have a particular semantic meaning, which will be important for our later proof: Any $(\boldsymbol{X}'', \boldsymbol{A}'') \in \mathbb{J}_{q_{\boldsymbol{X}}, q_{\boldsymbol{A}}}$ has $q_{\boldsymbol{X}}$ attribute bits and $q_{\boldsymbol{A}}$ adjacency bits that have the same value in $(\boldsymbol{X}, \boldsymbol{A})$, and a different value in $(\boldsymbol{X}', \boldsymbol{A}')$:

$$\left((\boldsymbol{X}'', \boldsymbol{A}'') \in \mathbb{J}_{q_{\boldsymbol{X}}, q_{\boldsymbol{A}}}\right) \iff$$
$$\left|\{m, d | X''_{m,d} = X_{m,d} \neq X'_{m,d}\}\right| = q_{\boldsymbol{X}} \wedge \left|\{i, j | A''_{i,j} = A_{i,j} \neq A'_{i,j}\}\right| = q_{\boldsymbol{A}}. \quad (52)$$

As proven by Lee et al. (2019), we can find an optimal solution to Eq. 48 by optimizing over the expected output of $\tilde{h}$ within each region of constant likelihood ratio. This can be implemented via the following linear program:

$$\Lambda_n\left(b_{\boldsymbol{X}_{\text{add}}}, b_{\boldsymbol{X}_{\text{del}}}, b_{\boldsymbol{A}_{\text{add}}}, b_{\boldsymbol{A}_{\text{del}}}, p_n,\right) :=$$

$$\min_{\boldsymbol{H}^{(n)}} \sum_{q_{\boldsymbol{X}}=0}^{\left(b_{\boldsymbol{X}_{\text{add}}} + b_{\boldsymbol{X}_{\text{del}}}\right)} \sum_{q_{\boldsymbol{A}}=0}^{\left(b_{\boldsymbol{A}_{\text{add}}} + b_{\boldsymbol{A}_{\text{del}}}\right)} H_{q_{\boldsymbol{X}}, q_{\boldsymbol{A}}}^{(n)} \Pr\left[\phi_{\text{attr}}(X'), \phi_{\text{adj}}(A')\right] \in \mathbb{J}_{q_{\boldsymbol{X}}, q_{\boldsymbol{A}}}) \quad (53)$$

$$\text{s.t.} \quad \sum_{q_{\boldsymbol{X}}=0}^{\left(b_{\boldsymbol{X}_{\text{add}}} + b_{\boldsymbol{X}_{\text{del}}}\right)} \sum_{q_{\boldsymbol{A}}=0}^{\left(b_{\boldsymbol{A}_{\text{add}}} + b_{\boldsymbol{A}_{\text{del}}}\right)} H_{q_{\boldsymbol{X}}, q_{\boldsymbol{A}}}^{(n)} \Pr([\phi_{\text{attr}}(X), \phi_{\text{adj}}(A)) \in \mathbb{J}_{q_{\boldsymbol{X}}, q_{\boldsymbol{A}}}] = p_n, \quad (54)$$

$$\boldsymbol{H}^{(n)} \in [0, 1]^{\left(r_{X_{\text{add}}} + r_{X_{\text{del}}}\right) \times \left(r_{A_{\text{add}}} + r_{A_{\text{del}}}\right)}. \quad (55)$$

Any optimal solution $\tilde{\boldsymbol{H}}^{(n)}$ corresponds to a single-output classifier $\tilde{h}_n$ that, given an input graph $(\boldsymbol{X}'', \boldsymbol{A}'')$, simply counts the number of attribute bits $q_{\boldsymbol{X}}$ and adjacency bits $q_{\boldsymbol{A}}$ that have the same

value in $(\boldsymbol{X}, \boldsymbol{A})$ and a different value in $(\boldsymbol{X}', \boldsymbol{A}')$ and then assigns a probability of $H_{q_{\boldsymbol{X}}, q_{\boldsymbol{A}}}^{(n)}$ to class $f_n$ and $1 - H_{q_{\boldsymbol{X}}, q_{\boldsymbol{A}}}^{(n)}$ to the remaining classes.

The optimal value of $Eq.$ 53 being larger than $0.5$ for a fixed perturbed graph $(\boldsymbol{X}', \boldsymbol{A}')$ only proofs that this particular graph is not a successful attack on $f_n$. For a robustness certificate, we want to know the result for a worst-case graph. However, the result is only dependent on the number of perturbations $b_{\boldsymbol{X}_{\text{add}}}, b_{\boldsymbol{X}_{\text{del}}}, b_{\boldsymbol{A}_{\text{add}}}$ and $b_{\boldsymbol{A}_{\text{del}}}$, and not on which specific bits are perturbed. Therefore, we can solve the problem for an arbitrary fixed perturbed graph with the given number of perturbations, and obtain a valid robustness certificate.

For use in our collective certificate, we define the set of budgets $\mathbb{K}^{(n)}$ for which prediction $f_n$ is certifiably robust as

$$\mathbb{K}^{(n)} = \left\{ \begin{pmatrix} b_{\boldsymbol{X}_{\text{add}}} & b_{\boldsymbol{X}_{\text{del}}} & b_{\boldsymbol{A}_{\text{add}}} & b_{\boldsymbol{A}_{\text{del}}} \end{pmatrix}^T \in \mathbb{L} \mid \Lambda_n \left( b_{\boldsymbol{X}_{\text{add}}}, b_{\boldsymbol{X}_{\text{del}}}, b_{\boldsymbol{A}_{\text{add}}}, b_{\boldsymbol{A}_{\text{del}}}, p_n \right) > 0.5 \right\} \quad (56)$$

where $\Lambda_n$ is defined as in Eq. 53 and $\mathbb{L} = [r_{\boldsymbol{X}_{\text{add}}}] \times [r_{\boldsymbol{X}_{\text{del}}}] \times [r_{\boldsymbol{A}_{\text{add}}}] \times [r_{\boldsymbol{A}_{\text{del}}}]$ (with $[k] = \{0, \ldots, k\}$) is the set of vectors that do not exceed the available collective budget (see Section 3).

### E.2 Tightness proof

With the definition of our base certificate in place, we can now formalize and prove that the resulting collective certificate is tight. Recall that randomized smoothing is a black-box method. The classifier that is being smoothed is treated as unknown. A robustness certificate based on randomized smoothing has to account for the worst-case (i.e. least robust under the given threat model) classifier. Our collective certificate lower-bounds the number of predictions that are guaranteed to be simultaneously robust. We show that with the randomized smoothing base certificate from the previous section, it actually yields the exact number of robust predictions, assuming the worst-case unsmoothed classifier.

**Theorem 1** *Let $(\boldsymbol{X}, \boldsymbol{A})$ be an unperturbed graph. Let $h : \mathbb{G} \mapsto \{1, \ldots, C\}^N$ be a (potentially non-deterministic) multi-output classifier. Let $f : \mathbb{G} \mapsto \{1, \ldots, C\}$ be the corresponding smoothed classifier with*

$$f_n(\boldsymbol{X}'', \boldsymbol{A}'') = \text{argmax}_{c \in \{1, \ldots, C\}} \Pr\left[ h_n(\phi_{\text{attr}}(\boldsymbol{X}''), \phi_{\text{adj}}(\boldsymbol{A}'')) = c \right], \quad (57)$$

$$y_n = f_n(\boldsymbol{X}, \boldsymbol{A}), \quad (58)$$

$$p_n = \Pr\left[ h_n(\phi_{\text{attr}}(\boldsymbol{X}), \phi_{\text{adj}}(\boldsymbol{A})) = y_n \right], \quad (59)$$

*and randomization schemes $\phi_{\text{attr}}(\boldsymbol{X}), \phi_{\text{adj}}(\boldsymbol{A})$ defined as in Eq. 45 and Eq. 46. Let $\boldsymbol{\psi} \in \{0, 1\}^N, \boldsymbol{\Psi} \in \{0, 1\}^{N \times N}$ be receptive field indicators corresponding to $f_n$ (see Eq. 2). Let $\mathbb{B}_\mathcal{G}$ be a set of admissible perturbed graphs, constrained by parameters $r_{\boldsymbol{X}_{\text{add}}}, r_{\boldsymbol{X}_{\text{del}}}, r_{\boldsymbol{A}_{\text{add}}}, r_{\boldsymbol{A}_{\text{del}}}, r_{\boldsymbol{X}_{\text{add,loc}}}, r_{\boldsymbol{X}_{\text{del,loc}}}, r_{\boldsymbol{A}_{\text{add,loc}}}, r_{\boldsymbol{A}_{\text{del,loc}}}, \sigma$, as defined in Section 2. Let $\mathbb{T}$ be the indices of nodes targeted by an adversary. Under the given parameters, let $o*$ be the optimal value of the optimization problem defined in Section C.*

*Then there are a perturbed graph $(\boldsymbol{X}', \boldsymbol{A}')$, a non-deterministic multi-output classifier $\tilde{h}$ and a corresponding smoothed multi-output classifier $\tilde{f}$ with*

$$\tilde{f}_n(\boldsymbol{X}, \boldsymbol{A}) = \text{argmax}_{c \in \{1, \ldots, C\}} \Pr\left[ \tilde{h}_n(\phi_{\text{attr}}(\boldsymbol{X}), \phi_{\text{adj}}(\boldsymbol{A})) = c \right] \quad \forall n \in \{1, \ldots, N\} \quad (60)$$

*such that*

$$\left| \left\{ n \in \mathbb{T} \mid \tilde{f}_n(\boldsymbol{X}', \boldsymbol{A}') = y_n \right\} \right| = o*, \quad (61)$$

$$\Pr\left[ \tilde{h}_n(\phi_{\text{attr}}(\boldsymbol{X}), \phi_{\text{adj}}(\boldsymbol{A})) = y_n \right] = p_n, \quad (62)$$

*and each $\tilde{f}_n$ is only dependent on nodes and edges for which $\boldsymbol{\psi}^{(n)}$ and $\boldsymbol{\Psi}^{(n)}$ have value 1.*

**Proof**: The optimization problem from Section C has three parameters $b_{\boldsymbol{X}_{\text{add}}}, b_{\boldsymbol{X}_{\text{del}}}, \boldsymbol{B}_{\boldsymbol{A}}$, which specify the budget allocation of the adversary. Let $b^*_{\boldsymbol{X}_{\text{add}}}, b^*_{\boldsymbol{X}_{\text{del}}}, \boldsymbol{B}^*_{\boldsymbol{A}}$ be their value in the optimum. We can construct a perturbed graph $(\boldsymbol{X}', \boldsymbol{A}')$ from the clean graph $(\boldsymbol{X}, \boldsymbol{A})$ as follows: For every node $n$, set the first $b^*_{\boldsymbol{X}_{\text{add}} n}$ zero-valued bits to one and the first $b_{\boldsymbol{X}_{\text{del}} n}^*$ non-zero bits to zero. Then,

flip any entry $(n, m)$ of $A_{n,m}$ for which $B^*_{A_{n,m}} = 1$. The parameters $b^*_{X_{\text{add}}}, b^*_{X_{\text{del}}}, B^*_A$ are part of a feasible solution to the optimization problem. In particular, they must fulfill constraints Eq. 31 to Eq. 36, which guarantee that the constructed graph is in $\mathbb{B}_\mathcal{G}$.

Given the perturbed graph $(X', A')$, we can calculate the amount of perturbation in the receptive field of each prediction $f_n$:

$$u^{(n)}_{X_{\text{add}}} = (b^*_{X_{\text{add}}})^T \psi^{(n)} \tag{63}$$

$$u^{(n)}_{X_{\text{del}}} = (b^*_{X_{\text{del}}})^T \psi^{(n)} \tag{64}$$

$$u^{(n)}_{A_{\text{add}}} = \sum_{(i,j):A_{i,j}=0} \Psi_{i,j} B^*_{A_{i,j}} \tag{65}$$

$$u^{(n)}_{A_{\text{add}}} = \sum_{(i,j):A_{i,j}=1} \Psi_{i,j} B^*_{A_{i,j}} \tag{66}$$

We can now specify the unsmoothed multi-output classifier $\tilde{h}$. Recall that in the collective certificate's optimization problem, each $f_n$ is associated with a binary variable $t_n$. In the optimum, $(t^*_n = 0) \iff \left[\left(u^{(n)}_{X_{\text{add}}} \quad u^{(n)}_{X_{\text{del}}} \quad u^{(n)}_{A_{\text{add}}} \quad u^{(n)}_{A_{\text{del}}}\right)^T \in \mathbb{K}^{(n)}\right]$, i.e. $f_n$'s robustness is guaranteed by the base certificate, and $o^* = \sum_{n \in \mathbb{T}} |\mathbb{T}| - t^*_n$.

**Case 1:** $n \notin \mathbb{T}$. Choose $\tilde{h}_n = h_n$. Trivially, constraint Eq. 62 is fulfilled since $\tilde{f}_n$ is only dependent on nodes and edges for which $\psi^{(n)}$ and $\Psi^{(n)}$ have value 1. Whether $\tilde{f}$ is adversarially attacked or not does not influence Eq. 61, as $n \notin \mathbb{T}$.

**Case 2:** $n \in \mathbb{T}$ and $t^*_n = 0$. Choose $\tilde{h}_n = h_n$. Again, constraint Eq. 62 is fulfilled since $\tilde{f}_n$ is only dependent on nodes and edges for which $\psi^{(n)}$ and $\Psi^{(n)}$ have value 1. Since $t^*_n = 0$, we know that $\left(u^{(n)}_{X_{\text{add}}} \quad u^{(n)}_{X_{\text{del}}} \quad u^{(n)}_{A_{\text{add}}} \quad u^{(n)}_{A_{\text{del}}}\right)^T \in \mathbb{K}^{(n)}$, i.e. $\tilde{f}_n(X', A') = y_n$.

**Case 3:** $n \in \mathbb{T}$ and $t^*_n = 1$. Since $t^*_n = 1$, we know that $f_n$ is not certified by the base certificate: $\left(u^{(n)}_{X_{\text{add}}} \quad u^{(n)}_{X_{\text{del}}} \quad u^{(n)}_{A_{\text{add}}} \quad u^{(n)}_{A_{\text{del}}}\right)^T \notin \mathbb{K}^{(n)}$. Let $H^{*(n)} \in [0,1]^{\left(u^{(n)}_{X_{\text{add}}} + u^{(n)}_{X_{\text{del}}}\right) \times \left(u^{(n)}_{A_{\text{add}}} + u^{(n)}_{A_{\text{del}}}\right)}$ be the optimum of the linear program underlying the base certificate (see Eq. 53 to Eq. 55). Define $\tilde{h}_n$ to have the following non-deterministic output behavior:

$$\Pr\left[\tilde{h}_n(X'', A'') = y_n\right] = H^{*(n)}_{q^{(n)}_X(X''), q^{(n)}_A(A'')} \quad \forall (X'', A'') \in \mathbb{G} \tag{67}$$

$$\Pr\left[\tilde{h}_n(X'', A'') = y'_n\right] = 1 - H^{*(n)}_{q^{(n)}_X(X''), q^{(n)}_A(A'')} \quad \forall (X'', A'') \in \mathbb{G} \tag{68}$$

for some $y'_n \neq y_n$ and with

$$q^{(n)}_X(X'') = \left|\{(n,d)|X''_{n,d} = X_{n,d} \neq X'_{n,d}\}\right| \tag{69}$$

$$q^{(n)}_A(A'') = \left|\{(n,m)|A''_{n,m} = A_{n,m} \neq A'_{n,m}\}\right|. \tag{70}$$

As discussed at the end of Section E.1, this classifier simply counts the number of bits in $(X'', A'')$ that are within $f_n$'s receptive field and have the same value in the clean graph $(X, A)$ and a different value in the perturbed graph $(X', A'')$. Since $H^{*(n)}$ is a valid solution to the linear program underlying the base certificate, we know that Eq. 62 is fulfilled, as it is equivalent to Eq. 54 from the base certificate. Since $\left(u^{(n)}_{X_{\text{add}}} \quad u^{(n)}_{X_{\text{del}}} \quad u^{(n)}_{A_{\text{add}}} \quad u^{(n)}_{A_{\text{del}}}\right)^T \notin \mathbb{K}^{(n)}$, we know that $\Pr\left[\tilde{h}_n(\phi_{\text{attr}}(X'), \phi_{\text{adj}}(A')) = y'_n\right] \geq 0.5$ (see Eq. 56), i.e. $\tilde{f}_n$ is successfully attacked, $\tilde{f}_n(X', A') = y'_n \neq \tilde{f}_n(X, A)$.

By construction, we have exactly $o^*$ nodes for which $\tilde{f}_n(X', A') = y_n$ and the remaining constraints are fulfilled as well. $\square$

# F  HYPERPARAMETERS

**Training schedule for smoothed classifiers.** Training is performed in a semi-supervised fashion with 20 nodes per class as a train set. Another 20 nodes per class serve as a validation set. Models are trained with Adam (learning rate = 0.001 [0.01 for SMA], $\beta_1 = 0.9$, $\beta_2 = 0.999$, $\epsilon = 10^{-8}$, weight decay = 0.001) for 3000 epochs, using the average cross-entropy loss across all training set nodes, with a batch size of 1. We employ early stopping, if the validation loss does not decrease for 50 epochs (300 epochs for SMA). In each epoch, a different graph is sampled from the smoothing distribution. We do not use the KL-divergence based regularization loss proposed for RGCN, as we found it to decrease the certifiable robustness of the model.

**Training schedule for non-smoothed GCN.** Training is performed in a semi-supervised fashion with 20 nodes per class as a train set. Another 20 nodes per class serve as a validation set. For the first 100 of 1000 epochs, models are trained with Adam (learning rate = 0.01, $\beta_1 = 0.9$, $\beta_2 = 0.999$, $\epsilon = 10^{-8}$, weight decay = $10^{-5}$), using the average cross-entropy loss across all training set nodes, with a batch size of 8. After 100 episodes, we add the robust loss proposed in (Zügner & Günnemann, 2019) (local budget $q = 21$, global budget $Q = 12$, training node classification margin = $\log(90/10)$, unlabeled node classification margin = $\log(60/40)$). The gradient for the robust loss term is accumulated over 5 epochs before each weight update in order to simulate larger batch sizes.

**Network parameters.** In all models, hidden linear and convolutional layers are followed by a ReLU nonlinearity. During training, each ReLU nonlinearity is followed by 50% dropout. For GCN and GAT, we use two convolution layers with 64 hidden activations. The number of attention heads for GAT is set to 8 for the first layer and 1 for the second layer. RGCN uses independent gaussians as its internal representation (i.e. each feature dimension has a mean and a variance). For RGCN, we use one linear layer, followed by two convolutional layers. We set the number of hidden activations to 32 for the means and 32 for the variances. Dropout is applied to both the means and variances. For APPNP, we use two linear layers with 64 hidden activations, followed by a propagation layer based on approximate personalized pagerank (teleport probablity = 0.15, iterations = 10). To ensure locality, we set all but the top 64 of each row in the approximate pagerank matrix to 0. For SMA, we first transform each node's features using a linear layer with 64 hidden activations. We then apply soft medoid aggregation ($k = 64$, $T = 10$) based on the approximate personalized pagerank matrix (teleport probablity = 0.15, iterations = 2). Note that we use the alternative parameterization from appendix B.5 of (Geisler et al., 2020) that is designed to improve robustness to attribute perturbations. After the aggregation, we apply a ReLU nonlinearity. Finally, we apply a second linear layer to each node independently.

**Randomized smoothing.** Randomized smoothing introduces four additional hyperparameters $\theta_{\boldsymbol{X}_{\text{add}}}, \theta_{\boldsymbol{X}_{\text{add}}}, \theta_{\boldsymbol{A}_{\text{add}}}, \theta_{\boldsymbol{A}_{\text{del}}}$, which control the probability of flipping bits in the attribute and adjacency matrix under the smoothing distribution. If we only certify attribute perturbations, we set $\theta_{\boldsymbol{A}_{\text{add}}} = \theta_{\boldsymbol{A}_{\text{del}}} = 0$, $\theta_{\boldsymbol{X}_{\text{add}}} = 0.002$ and $\theta_{\boldsymbol{X}_{\text{del}}} = 0.6$. If we only certify adjacency perturbations, we set $\theta_{\boldsymbol{X}_{\text{add}}} = \theta_{\boldsymbol{X}_{\text{del}}} = 0$, $\theta_{\boldsymbol{A}_{\text{add}}} = 0$ and $\theta_{\boldsymbol{A}_{\text{del}}} = 0.4$. If we jointly certify attribute and adjacency perturbations, we set $\theta_{\boldsymbol{X}_{\text{add}}} = 0.002, \theta_{\boldsymbol{X}_{\text{del}}} = 0.6, \theta_{\boldsymbol{A}_{\text{add}}} = 0$ and $\theta_{\boldsymbol{A}_{\text{del}}} = 0.4$. Exactly evaluating smoothed classifiers is not possible, they have to be approximated via sampling. We use 1000 samples to determine a classifier's majority class ($y_n$), followed by $10^6$ samples to estimate the probability of the majority class ($p_n$) via a Clopper-Pearson confidence interval. Applying Bonferroni correction, the confidence level for each confidence interval is set to $1 - 0.01/N$ to obtain an overall confidence level of 99% for all certificates.

