# OpenReview forum: "Collective Robustness Certificates: Exploiting Interdependence in Graph Neural Networks"
_ICLR.cc/2021/Conference — ICLR 2021 Poster_

### Official Review · AnonReviewer1 · 2020-10-27
**great work but some comparisons could be missing.**

**Rating:** 8
**Confidence:** 2

**Review:**

Summary
-------------
Current methods on adversarial robustness certificates consider data points independently which are highly pessimistic for structured data. This work proposes the first collective robustness certificate that considers the structure of the graph by modeling locality in order to derive stronger guarantees  that the predictions remain stable under perturbations.

This work focuses on Graph Neural Networks comparing between a Naive collective certificate (baseline) and a proposed collective certificate that combines single-node certificates effectively.

The experiments compares these two methods against certified ratio vs. attribute and edge perturbations on the datasets Cora-ML, Citeseer, and PubMed.

Pros
------
- The paper is well-written and easy to follow.
- The paper tries to address a very common problem of adversarial attacks where data points are structured. Although it is a common problem, it was not explored with respect to collective robustness certificates before this work.
- The paper shows a novel, effective way of combining individual certificates by incorporating locality.
- The paper presents an LP-relaxation method that allows us to solve the certificate fast for large graphs where mixed-integer problems are prohibitively costly.
- The paper shows strong theory and experiments to illustrate the efficacy of the proposed collective certificate.
- The experiments show run-time and uncertainty measures with multiple runs for statistical significance.

Questions
--------------
- Is there a reason why [1] was not discussed in the paper? it is highly relevant as it also studies adversarial robustness for structural attacks.
- How does the proposed method apply to robust Associative Markov Networks (AMN) in [1]?
- Did you try running the proposed method on WebKB and Reuters which are present in this work [1]?
- How does the linear relaxation in this work differ from the one provided in [1]?
- How would the proposed method compare against robust AMN [1]?
- Can the findings be used to come up with robust methods outside classification, like segmentation and scene understanding?

In summary, I like the novelty of this method and the through experiments that were conducted that illustrate the efficacy of the proposed collective certificate, thus I recommend an accept.

[1] Kai Zhou, Yevgeniy Vorobeychik. Robust Collective Classification against Structural Attacks. UAI 2020.

------- Post rebuttal
I am satisfied with most of the rebuttal the authors have provided, and I have raised my score to an 8.

---

> ### Author Response · Authors · 2020-11-18
> **Response - Reviewer 1**
>
> Thank you for your review!
>
> Before responding to your questions, let us briefly summarize the mentioned paper ([1]) for other readers. The paper deals with the adversarial robustness of pairwise associative Markov networks (AMN), a type of probabilistic graphical model for node classification. The authors propose a robust loss function that maximizes the margin between the likelihood of the ground-truth labels and that of all other possible labels under adversarial perturbation.
>
> ### Responses
> **Is there a reason why [1] was not discussed in the paper?**
> The robust loss function in [1] is not a robustness certificate. While it compares favorably to standard training under different adversarial attacks, it does not provide any provable guarantees. In our original submission we simply refer to a survey paper of non-provable defenses that have subsequently been broken by novel adversarial attacks. Based on your review we have updated our manuscript to include an overview of different adversarial defenses (including [1]) , which will hopefully allow for a better differentiation between robustness certificates and defenses that do not provide provable guarantees.
>
> **How does the proposed method apply to robust Associative Markov Networks (AMN) in [1]?**
> The proposed method can in principle be applied to AMNs. All that is needed is some base certification procedure that can guarantee the robustness of individual predictions to adversarial attacks. The many base certificates can then be combined into a collective certificate using our method.
>
> **Did you try running the proposed method on WebKB and Reuters which are present in this work [1]?**
> Based on your comment, we have run additional experiments on graphs constructed from the WebKB and Reuters corpora (see Fig. 8). Since we could not find an official reference implementation, the constructed graphs might be slightly different from those in [1] due to random sampling, tie-breaking among nodes with equal cosine similarity, etc.
> Further note that the multiple classes are not merged into two super-classes. Repeating the experiment with binary class labels as done in [1] yielded results even slightly better than the multi-class results shown on Fig. 8.
>
> **How does the linear relaxation in this work differ from the one provided in [1]?**
> This work and [1] propose two different mixed-integer programs, one minimizing the number of robust predictions, the other one minimizing a likelihood margin. In both cases the integer variables are relaxed to real-valued variables. Furthermore both mixed-integer programs involve boolean logic that is expressed using linear constraints. Quoting section 3.2 of [1], these are “standard techniques” and not the core contribution of either paper.
>
> **How would the proposed method compare against robust AMN [1]?**
> As discussed in the response to the first question, the proposed method is a robustness certificate that provides provable guarantees while the method from [1] is not.
> While robustness certificates can be evaluated based on their certified ratio, evaluating other defenses requires powerful adversarial attacks that are specifically adapted to break them.
>
> **Can the findings be used to come up with robust methods outside classification, like segmentation and scene understanding?**
> The proposed method can in principle be applied to any task in which many labels are predicted for a single shared input, including segmentation and scene understanding. The performance gain over the naive certificate will depend on the degree of locality of the classifier architecture.
>
>
>
> [1] Kai Zhou, Yevgeniy Vorobeychik. Robust Collective Classification against Structural Attacks. UAI 2020.

---

### Official Review · AnonReviewer2 · 2020-10-27
**This paper first proposes a collective robustness certificate by fusing individual certificates into a provably stronger one, which significantly outperforms existing adversial certificates. Thus, I vote for accepantance.**

**Rating:** 6
**Confidence:** 1

**Review:**

This paper addresses the limitation of the existing adversarial robustness certificates that ignores that a single shared input is present, and thus assumes an adversary can use different perturbed inputs to attack different predictions. A novel collective certificate fusing single certificates into a stronger one, is proposed by explicitly modeling local structure of input data using graph convolution node classifiers. In terms of certified ratio, the collective certificate significantly improve the results compared with existing individual certificates.

-quality: the technical quality is sound.

-clarity: the input data, problem formulation and method are clearly described.

-originality & significance: it is the first attempt in considering collective robust certificates (CRCs) by fusing individual adversarial certificate. As shown in the experiments, the certified ratio of the CRC is significantly improved over existing adversarial one. I think the collective robust certificate has some impacts for robust graph node classifications.

Pros: The paper is well motivated. The problem and the method are both clearly presented. The improvements of the collective robust certificate over the existing ones is sufficiently high in terms of certified ratios.

Cons: I do not find any notable weaknesses.

---

> ### Author Response · Authors · 2020-11-18
> **Response - Reviewer 2**
>
> Thank you for your review!
>
> We are pleased to hear that you did not find any notable weaknesses and that you are convinced by the overall quality of the paper’s writing and content.

---

### Official Review · AnonReviewer3 · 2020-10-28
**Interesting paper. Well-motivated. Good results**

**Rating:** 7
**Confidence:** 3

**Review:**

** Summary:
In the context of structured prediction, where multiple predictions are simultaneously made based on a single input, this works argue that existing robustness certificates independently operating on each node prediction end up with overly pessimistic results. Rather than that, this work advocates to collectively certify the overall accuracy using a single perturbed graph at a time. Starting from the building-blocks of base certificates, the authors formulate a global optimization problem, which is made tractable via a number of relaxation steps resulting in a final mixed-integer linear programming (MILP). Experimental results demonstrate clear advantage of the proposed certificates over base ones, with reasonable computational overheads coming from solving the MILP.

The paper is well-motivated, well-written and easy to follow. I think this is a valid method to assess robustness of classifier satisfying locality like GCN.

** Strength:
 - This work is well-motivated. The arguments are valid on the limitations of independent based certificates for collective tasks. Experimental results convincingly show how such certificates are pessimistic in the addressed context.
 - The writing is clear and well-structured, easy to understand and follow.
 - Nice discussion on the limitations

** Limitations:
 - Typos:
 	+ Eqn. (2): $f_n(\boldsymbol {X}^{'}, \boldsymbol{A}^{'}) = f_n(\boldsymbol{X}^{''}, \boldsymbol{A}^{ \textcolor{red}{''}})$

 	+ Eqn. (7): $\boldsymbol {X}^{''}_{i,d} = \psi_i^{(n)}\boldsymbol {\textcolor{red}{X}}^{'}_{i,d} + (1-\psi_i^{(n)})\boldsymbol {\textcolor{red}{X}}_{i,d}$



** Justification of rating: overall this is an interesting paper. The motivation, arguments and results are convincing.

---

> ### Author Response · Authors · 2020-11-18
> **Response - Reviewer 3**
>
> Thank you for your review!
>
> We are glad to know that you found the paper well-written, the work well-motivated and the results convincing.
> We have corrected the two typos you pointed out in the updated version of the manuscript.

---

### Official Review · AnonReviewer4 · 2020-10-29
**This paper proposes a new concept called “collective robustness certificate” that computes the number of predictions which are simultaneously guaranteed to remain stable under perturbation.**

**Rating:** 5
**Confidence:** 1

**Review:**

This paper studies classifiers that collectively output many predictions based on a single input. Existing adversarial robustness certificates assume that an adversary can use different perturbed inputs to attack different predictions, and ignore the fact of a single shared input, thereby being overly pessimistic. This paper proposes a collective certificate that computes the number of simultaneously certifiable nodes for which the predictions can be guaranteed to be stable (not change). It is conducted basically by fusing individual certificates into a provably stronger certificate through explicitly modeling locality.
Pros: This is the first effort that considers collective robustness certificate.
Cons:
1.	As discussed in the paper, the proposed approach is designed to exploit locality. Without locality, it is equivalent to a naïve combination of base certificates that sums over perturbations in the entire graph.
2.	The writing of the paper can be improved. The abstract seems to be unfinished. It appears to be hard to include sufficient preliminaries to clearly describe the research problem in a conference paper. It’s probably better to have a longer version as a journal paper.

---

> ### Author Response · Authors · 2020-11-18
> **Response - Reviewer 4**
>
> Thank you for your review!
>
> **Concerning 1.):**
> As you correctly pointed out, the locality assumption is made transparent to the reader -- the manuscript includes an entire section dedicated to discussing the limitations. Nonetheless, popular GNN architectures satisfy locality in practice, which results in a significant increase in the certified ratio using our method, as shown in our experiments. This limitation can be alleviated in future work but that is is out of scope for this paper.
>
> **Concerning 2.):**
> We believe that the paper is sufficiently self-contained and provides all preliminaries needed to understand the research problem. However, if you have any specific questions we would be glad to answer them and adapt the manuscript to resolve any unclarities.

---

### Author Response · Authors · 2020-11-18
**Summary / Changelog**

This post serves as a summary of updates since the initial submission of our manuscript.

We have replied to all reviewers and made the following changes in response to their comments:
* Reviewer 1:
  * Add overview over different heuristic defenses for collective tasks
  * Add experiments on WebKB and Reuters dataset (Appendix A)
* Reviewer 3:
  * Correct typos in Eq.2 and Eq.7

We have further made the following minor changes:
* correct caption of Fig. 8 (Cora, Citeseer, Pubmed -> WebKB, Reuters)
* remove unnecessary "through" at end of second paragraph
* correct indentation in Algorithm 2
* correct arXiv references (bibliography style does not support eprint field)
* increase x-lim and y-lim on Fig. 3
* correct indexing in Eq. 75 and Eq. 76 (h_n instead of h)

---

### Decision · Program_Chairs · 2021-01-07
**Final Decision**

**Decision:**

Accept (Poster)

**Comment:**

This paper considers a new setting of robustness, where multiple predictions are simultaneously made based on a single input. Different from existing robustness certificates which independently consider perturbation of each prediction, the authors propose collective robustness certificate that computes the number of predictions which are simultaneously guaranteed to remain stable under perturbation. This yields more optimistic results. Most reviewers think this is a very interesting work and the authors present an effective method to combine individual certificate. The experimental results are convincing. I recommend accept.